

# Subseasonal hydrometeorological ensemble predictions in small- and medium-size mountainous catchments: Benefits of the NWP approach

Samuel Monhart[1,2,3], Massimiliano Zappa[1], Christoph Spirig[2], Christoph Schär[3], and Konrad Bogner[1]

[1]Swiss Federal Institute for Forest, Snow and Landscape Research WSL, Mountain Hydrology and Mass Movements, Birmensdorf, Switzerland
[2]Federal Office of Meteorology and Climatology MeteoSwiss, Climate Prediction, Zurich-Airport, Switzerland
[3]ETH Zurich, Institute for Atmospheric and Climate Science, Zurich, Switzerland

*Correspondence to*: Samuel Monhart (Samuel.Monhart@wsl.ch)

**Abstract.** Traditional Ensemble Streamflow Prediction systems (ESP) are known to provide a valuable baseline to predict streamflows at the subseasonal to seasonal timescale. They exploit a combination of initial conditions and past meteorological observations, and can often provide useful forecasts of the expected streamflow in the upcoming month. In recent years, numerical weather prediction (NWP) models for subseasonal to seasonal timescales have made large progress and can provide added value to such a traditional ESP approach. Prior of using such meteorological predictions two major problems need to be solved: the correction of biases, and downscaling to account to increase the spatial resolution. Various methods exist to overcome these problems, but the potential of using NWP information and the relative merit of the different statistical and modeling steps remains open. To address this question, we compare a traditional ESP system with a subseasonal hydrometeorological ensemble prediction system in three alpine catchments with varying hydroclimatic conditions with areas between 80 and 1700 km². Uncorrected and corrected (pre-processed) temperature and precipitation reforecasts from the ECMWF subseasonal NWP model are used to run the hydrological simulations and the performance of the resulting streamflow predictions is assessed with commonly used verification scores characterizing different aspects of the forecasts (ensemble mean and spread). Our results indicate that the NWP based approach can provide superior prediction than the ESP approach, especially at shorter lead times. In snow-dominated catchments the pre-processing of the meteorological input further improves the performance of the predictions. This is most pronounced in late winter and spring when snow melting occurs. Moreover, our results highlight the importance of snow related processes for subseasonal streamflow predictions in mountainous regions.

## 1 Introduction

Subseasonal hydrometeorological predictions are of special interest for many different applications in the public and the private sectors. For example to develop early warning systems for flood and drought preparedness for the general public (Bogner et al., 2018; Cloke and Pappenberger, 2009; Fundel et al., 2013a; Van Lanen et al., 2016; Schär et al., 2004; White et





al., 2017), to optimize the production of renewable energy sources such as wind (Beerli et al., 2017), solar (Inman et al., 2013) and hydropower (García-Morales and Dubus, 2007) or to ensure inland waterway transportation (Meißner et al., 2017).

In recent years the numerical weather prediction (NWP) systems have greatly evolved (Bauer et al., 2015). The ensemble forecasting approach introduced two decades ago allows capturing the chaotic nature of the atmosphere in a probabilistic sense and has extended the horizon to which weather predictions can provide valuable information. Hydrological prediction systems, especially beyond the short-range lead times, benefit from such an ensemble approach (Demargne et al., 2014; Jaun et al., 2008; Schaake et al., 2007; Verbunt et al., 2007). Ensemble hydrometeorological end-to-end prediction systems for the subseasonal time scale, i.e. forecasts for lead times up to 4 to 6 weeks, are now being developed and investigated for different parts in the world. Nowadays, different research initiatives (Hao et al., 2018; Robertson et al., 2015; Vitart et al., 2017; Vitart and Robertson, 2018) set their focus on the assessment of predictability within this lead-time horizon. But still this lead time between the medium-range forecasts and the seasonal predictions is a grey zone in both, the meteorological and hydrological forecasting communities.

However, ensemble prediction systems have extensively been used for short to medium-range forecasts and it could be shown the resolution of the model plays an important role for a good performance of ensemble precipitation forecasts (Marsigli et al., 2008; Montani et al., 2001, 2003). A higher resolution can be achieved by dynamically downscaling the ensemble predictions, what led to a gain in predictability in ensemble precipitation forecast over the past years in the medium-range time scale (Montani et al., 2011). From a hydrological perspective flood peaks can much better be predicted using mesoscale ensemble forecast models compared to global models (Davolio et al., 2012). In such operational short to medium-range forecasting systems, dynamical downscaling methods are a valuable tool to increase the resolution of the forecasts but for long term predictions the computational demand of dynamical downscaling methods is a limiting factor and it has been shown that similar results can be achieved with statistical downscaling methods (Díez et al., 2005; Manzanas et al., 2018). In recent years, seasonal ensemble prediction systems have increasingly been used in the hydrological forecasting context. At this time scale downscaling is usually included in statistical processing techniques that primarily aim at correcting the bias of the meteorological forecast. Different statistical bias correction techniques can be used to improve the skill of seasonal forecasts (Bohn et al., 2010; Crochemore et al., 2016; Kumar et al., 2014). An alternative approach for seasonal streamflow prediction is the traditional ensemble prediction system (ESP) first introduced by Day (1985). In this approach, observed historical meteorological conditions are used to generate the streamflow forecasts. Several studies have shown that Ensemble Streamflow prediction (ESP) can provide skillful seasonal streamflow prediction (Harrigan et al., 2017) and seasonal hydrometeorological prediction systems have difficulties to outperform the ESP beyond one month lead time (Arnal et al., 2018; Lucatero et al., 2018). Irrespective of the methodology used, the performance of ensemble streamflow forecasts depend to a large degree on the initial condition within the catchment especially for small catchments (Van Dijk et al., 2013; Thirel et al., 2010). In subseasonal to seasonal hydrometeorological predictions, the memory of the river networks is predominately driven by the initial conditions of the land surface characteristics, i.e. soil moisture and snow cover (Jörg-Hess et al., 2015a). It has been shown that these parameters play an important role for skillful hydrometeorological forecasts using numerical weather



prediction (NWP) inputs in hydrological models for streamflow forecasting (Orth and Seneviratne, 2013b; Sinha and Sankarasubramanian, 2013). However, for small snow dominated catchments the benefit of using pre-processed subseasonal NWP input has not yet been investigated. The three main reasons for the lack of studies at smaller scales is the fact that the driving meteorological models are calculated on a global scale with coarse resolution. Therefore, small catchments are often

smaller than the nominal resolution of these models, making a pre-processing step necessary to bias correct and downscale the meteorological forecasts to an adequate spatial resolution as an input to the hydrological models. The second reason is that small catchments usually do not have a long temporal memory and processes leading to streamflows are rather quick and therefore the skill in early lead times do not extend into longer lead times (Orth and Seneviratne, 2013a). As a consequence, the upper limits of the forecast skill at the subseasonal time scale strongly depend on such catchment characteristics (Bogner

et al., 2016, 2018). Finally, previous generations of subseasonal to seasonal climate forecasts rapidly lost skill beyond the first 2 weeks (see for example Lavers et al., 2009).

Small and medium sized catchments in mountainous regions such as the Alps are often snow dominated, hence the streamflow, and especially the forecasts thereof, depends to a large degree to snowmelt processes that are driven by temperature (Hock, 2003; Ohmura, 2001; Zappa et al., 2003). Monhart et al. (2018) have shown that the statistically corrected subseasonal

temperature forecasts exhibit positive skill compared to the climatological reference of up to 3 three weeks, whereas positive skill of precipitation forecasts is restricted to the first 10 days. Hence the question arises if and to what extent the positive skill of temperature forecasts in meteorological prediction models does further progress to the streamflow forecasts. Skillful streamflow forecasts might not directly be related to the skill in temperature forecasts but rather in an appropriate sampling of the initial conditions and the actual snow cover (snow water equivalent) at initialization (Jörg-Hess et al., 2015a). Therefore,

the skill of the streamflow forecasts will not solely depend on the quality of temperature forecasts alone. To investigate this question, ensemble streamflow prediction can be used to evaluate the importance of the initial conditions. The comparison of the skill of an ESP forecast and the skill of actual hydrometeorological predictions can indicate the relative importance of using physically consistent meteorological forecasts to produce skillful streamflow predictions.

**2. Methods**

The conceptual framework used in this study is presented in Figure 1. The hydrological model is run with different meteorological forcings to provide ensemble streamflow predictions. A traditional ESP approach using 34 years of meteorological observations provides the baseline forecasts. A second input is based on the reforecasts from the ECMWF subseasonal prediction system as described in section 3.1. Along this chain four different configurations are used to feed the

hydrological model and generate streamflow predictions: the raw reforecasts for both temperature and precipitation, the raw precipitation reforecast and pre-processed temperature reforecasts and vice versa, and both parameters pre-processed.



## 2.1 Pre-processing in the hydrometeorological model chain

In the pre-processing step a quantile mapping (QM) technique is applied in a leave-one-year-out setup to correct each reforecast year separately. The corrections are applied in a lead-time dependent manner. A description of the pre-processing setup can be found in Monhart et al. (2018). In contrast to the site-specific pre-processing setup in that study, a gridded observational product is used here. The pre-processing is performed for temperature and precipitation. The QM technique is a simple and widely used method for pre-processing hydrometeorological forecasts (e.g. Kang et al., 2010; Lucatero et al., 2018; Verkade et al., 2013). For a given target day of a reforecast the correction is derived from the distribution of all the reforecasts within a three weeks window around the same lead day and the corresponding observations, hence the correction depends both on the lead time and on the period of the years. For temperature an additive correction and for precipitation a multiplicative correction is applied.

## 2.2 Ensemble Streamflow Prediction (ESP)

The Ensemble streamflow predictions follow the established procedure first proposed by Day (1985). Many studies have shown the potential of this method to provide skillful streamflow predictions at the subseasonal to seasonal time scale in Europe (e.g. Arnal et al., 2018; Harrigan et al., 2017). The basic principle behind the ESP is to create an ensemble of streamflows based on known initial conditions and forced by historic climate sequences. The historical record used in this study covers the period from 1980 to 2014 resulting in 34 members for each forecast. As in the hydrometeorological model chain, the ESP approach is set up in a leave one year out manner to ensure that the information of the year to be verified is not part of the forecasts itself. ESP predictions can be skillful especially in catchments where the predictability is mainly driven by the initial conditions, although the quality of the predictions depends on the seasons and hydroclimatic characteristics (Wood and Lettenmaier, 2008).

## 2.3 Hydrological Simulations

The hydrological simulations are performed with the Precipitation-Runoff-Evapotranspiration-Hydrotope model (PREVAH) (Gurtz et al., 1999; Viviroli et al., 2009a; Zappa et al., 2003). In this study, the distributed model version which requires gridded input data is used as described in Speich et al. (2015) and first applied in a hydrological study by Schattan et al. (2013). The model requires spatial information (land use, aspect and elevation) and gridded meteorological variables. Besides the statistically corrected temperature and precipitation predictions, relative humidity, sunshine duration, surface albedo and solar radiation are needed to run PREVAH. For the initialization of the model the required fields (i.e. soil moisture, groundwater storages, snow cover) are used from a reference simulation driven with the gridded meteorological observation dataset as described in subsection 3.4. The distributed hydrological model is run at a resolution of 200 m x 200 m, hence the same model internal procedures are used to further downscale the meteorological inputs for all different experiments. Namely, an interpolation based on inverse distance weighting (IDW) and different height and terrain specific correction are applied to the


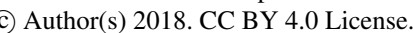


input variables (adiabatic lapse rate correction; aspect and slope corrections) as described in Zappa et al. (2003). A more extensive description of the model and a comparison of to the HRU based version of the model can be found in (Jörg-Hess et al., 2015a).

## 2.4 Verification

To verify the streamflow predictions various metrics are used to assess the forecast performance according to different characteristics or attributes of the forecasts respectively. For the selection of verifications metrics we follow the procedures presented in Brown et al. (2010) and we combine deterministic and probabilistic measures of skill to assess the forecast performance. More detailed reviews of ensemble forecast verification can be found in Jolliffe and Stephenson (2012) and Wilks (2006). Besides the Nash-Sutcliffe efficiency (NSE) (Nash & Sutcliffe, 1970) widely used to assess the performance of

hydrological models and the logarithmic version thereof (NSE log), which is more sensitive to low flows (Krause et al., 2005), we use the ensemble mean error (bias) and the mean absolute error (MAE). Although these metrics can describe the average characteristics of the ensemble forecasts it has been shown that it is crucial to consider the spread of the ensemble forecast as well to properly asses the forecast performance in particular at the subseasonal to seasonal time scale (Kumar et al., 2014). Therefore the continuous ranked probability skill score (CRPSS) with the extension proposed by Ferro (2014) to account for

small ensemble sizes is used to characterize the performance of the full ensemble (Hersbach, 2000; Müller et al., 2005). In addition, we use the spread to error ratio (SprErr) as an indicator for the forecast reliability (Hopson, 2014) and finally rank histograms to visually examine the reliability of the forecasts (Anderson, 1996; Talagrand et al., 1997; Hamill & Colucci, 1997). In each of these scores the reforecasts are compared to observations (ME and MAE) and the climatology of the reference simulation is used as a reference climatology for NSE and CRPSS. In this study we use pseudo observations from a reference

simulation to substitute real observations for the streamflow forecast verification. This is done to separate the effect on the performance of the hydrological predictions resulting from either different meteorological input forecasts or from the deficits in the hydrological model to simulate low flows. The setup of the reference simulation to generate the pseudo observations is described in the data subsection 3.4.

## 3. Data

### 3.1 Meteorological reforecast data

We obtained the subseasonal reforecasts from the ECMWF Integrated Forecasting System (IFS) version CY40r1. This version was operational from 19 November 2013 to 12 May 2015. This is a unique dataset because no system change occurred for nearly one and a half year (http://www.ecmwf.int/en/forecasts/documentation-and-support/changes-ecmwf-model/cycle-40r1/cycle-40r1 for the documentation of IFS CY40r1). Routinely the model is updated more frequently within one year and

therefore changes in the system, e.g. changes to the horizontal and vertical resolution or changes in the parametrization of physical processes affect the skill of the forecast over the course of the year. The same data set is used as in the study presented



by Monhart et al. (2018) and can be referred to for a more detailed description of the forecast system and for an extensive verification of these meteorological forecasts.

In this study the historical 5-member reforecasts (i.e. hindcasts) are used to drive the hydrological simulations. These meteorological reforecasts are run for all forecasts issued on Thursdays using ERA-interim analysis for the initialization (ECMWF, 2014) covering the period from April 1994 to March 2014. These reforecasts are essential for the post-processing of the meteorological forecasts (i.e. pre-processing from a hydrological point of view). The bias of the forecasting system can be estimated using the reforecasts and future forecasts can be corrected (or as in this analysis the reforecasts itself).

## 3.2 Meteorological observational data

We use gridded observation data sets at 2 km x 2 km resolution for daily temperature and precipitation to pre-process the meteorological reforecasts and for the verification thereof (Frei, 2014; Isotta et al., 2014; Meteoswiss, 2016; MeteoSwiss, 2016). These products are based on surface observations and are often used in climatological studies for Switzerland (e.g. Addor et al., 2016; Begert and Frei, 2018, Orth et al. (2015) for a hydrological model study with the same model version we use in this study). Nevertheless such gridded observational products exhibit limitations especially in complex terrain with high mountain peaks because of the interpolation errors and errors induced by natural variability (Frei, 2014; Addor & Fischer, 2015). Despite these limitations the analysis benefits by using the gridded version because of the scarcity of meteorological stations available in the catchment areas. The additional meteorological variables (relative humidity, sunshine duration, surface albedo and solar radiation) needed to run the hydrological model PREVAH are directly used from the (raw) meteorological forecasts and are inherently downscaled with the PREVAH internal methodology. Thus, no observations are needed for these variables. In addition, these variables are of minor importance for the forecast time scales investigated in this study.

## 3.3 Catchment characteristics and hydrological data

The experiments analyzed in this study are performed for three alpine catchments with various hydro-climatic characteristics. This allows identifying the underlying processes that lead to skillful subseasonal streamflow predictions. An overview of the catchment characteristics can be found in Table 1.

The Verzasca catchment located in the Southern part of the Alps is snow dominated in the cold seasons and more precipitation dominated in warm seasons (Wöhling et al., 2006). The average height of the catchment is 1651 m a.s.l. with a maximum height of 2864 m a.s.l. and a minimum height at the gauging station of 490 m a.s.l.. Despite this height difference the catchment is of medium size only with a total of 185 km$^2$. The runoff in this catchment is low in winter due to accumulation of snow in elevated regions and highly dynamical in late spring to early autumn because the area is prone to heavy thunderstorm activity (Bogner et al., 2018; Liechti et al., 2013). The data used for calibration of the hydrological model and for the verification of the reference simulation is provided by the Federal Office for the Environment (FOEN) for the gauging station Lavertezzo.

The Klöntal catchment, located in the Northern part of the Alps, is mainly snow dominated and inherits a glaciated area of about 3 km$^2$ ($< 5\%$). The maximum height is 2883m a.s.l. and the minimum height, corresponding to the height of lake Klöntal,



is 847 m a.s.l. with a catchment area 83 km$^2$ categorized as a small catchment. The lake is used for hydropower generation. The data provided by the hydropower operator is used for calibration of the hydrological model and the verification of the streamflow reforecasts. Due to the absence of a gauging station, this data is an estimate of the streamflow at the lake outlet and deduced from the water balance of historical lake levels. Hence, the quality of the data is lower compared to the other two

catchments and therefore the interpretation especially under low flow condition should be treated with care.

The third catchment under investigation is the pre-alpine Thur catchment. This catchment is of medium size with 1696 km$^2$ and mainly precipitation dominated. The highest elevation is 2505 meter, the lowest elevation at the gauging station in Andelfingen provided by the FOEN is at 356 m, and the mean catchment height is 770 m a.s.l.. This catchment has frequently been analyzed in literature (Bogner et al., 2016; Fundel et al., 2013b; Jörg-Hess et al., 2015b).

### 3.4 Hydrological pseudo observations: the reference simulation

For all three catchments described above, the reference simulation is generated using an observed gridded meteorological data set for the meteorological variables (temperature, precipitation, relative humidity, sunshine duration, surface albedo and solar radiation) to run the hydrological model PREVAH (see section 2.3) in the same setting as used for different previous studies

(Orth et al., 2015; Schattan et al., 2013; Speich et al., 2015). The gridding of all the meteorological variables makes use of elevation-based de-trending and inverse distance based interpolation as detailed in the baseline publication on PREVAH and its tools (Viviroli et al., 2009a) and also used for application of other hydrological models (e.g. Melsen et al., 2016). The reference simulation is a single time series starting in 1992 (after a spin up period of three years) and ending in 2015. The spin up period ensures the stability of the streamflow simulations and is of particular importance to fill low frequency storages for

baseflow and snow (Viviroli et al., 2009b). In addition the spin-up times highly vary across catchments depending on the hydroclimatic characteristics and catchment size (Rahman et al., 2016; Seck et al., 2015). The climatology of the reference simulation is referred to as reference climatology and is used as reference for the computation of skill scores.

### 4. Results

### 4.1 Performance of the reference simulation

The reference simulation is analysed over the full 20-year period of the reforecasts. The verification of the reference simulation against observations, summarized in Table 2, generally shows high agreement of the simulation with the corresponding observations.

The Nash-Sutcliffe efficiency coefficient (NSE) for the entire analysis period (FullYear) exhibits values above 0.84 for all catchments. Best performance can be observed in spring and lowest performance in winter when low flows dominate. The

logarithmic form of the NSE (NSE log) shows a similar picture with better performance in the Thur catchment. Some difficulties occur in DJF in the Verzasca catchment and in DJF and SON in the Klöntal catchment. In the Verzasca and the

Klöntal catchment the MAE and the absolute bias are constant over the course of the year except in winter. But compared to the mean annual flows of the specific catchments largest relative biases are observed during DJF. The Thur catchment exhibits smallest relative biases except in JJA when low flows occur because of the absence of snow in this catchment. Various applications do not focus on low flows but rather on flow volumes, e.g. hydropower operations are more interested in forecasts about the total upcoming flow volume to adapt and optimize their production. Therefore, we here focus on the standard Nash-Sutcliffe coefficient instead of the logarithmic form thereof.

We hereafter verify streamflow predictions against the reference simulation to focus on analyzing the effect of the different meteorological input forecasts. This allows to solely assess the effect of the pre-processing on the hydrological predictions by supressing the hydrological errors itself and is often done to evaluate of operational forecasting systems (Alfieri et al., 2014), to assess the contributions of uncertainty on the hydrological forecasts (Voisin et al., 2011), or for a comparison of the skill of different forecasting systems (Pappenberger et al., 2015). Such an evaluation against a reference simulation also minimizes the influence of the deficits of the hydrological model under low flow conditions on the verification analysis as found for the reference simulation described above. However, the effect of pre-processing on the performance if real observations are used for the verification reveals interesting aspects of the deficits of the hydrological simulations and will be discussed in section 5.

### 4.2 Skill of the meteorological input

The raw and bias corrected temperature and precipitation reforecasts used to drive the hydrological predictions are verified against the gridded observations in the Verzasca domain. In Figure 3, the CRPSS for the raw and the bias corrected temperature and precipitation reforecasts is shown. Raw temperature reforecasts mainly exhibit negative CRPSS values, indicating essentially no skill compared to climatology. After bias correction using the quantile mapping technique the reforecasts exhibit positive CRPSS up to 18-24 lead days in all seasons but spring (MAM). Raw precipitation reforecasts exhibit positive skill for lead days 5-11 in all seasons and negative CRPSS for longer lead times. After bias correction the skill is higher with positive CRPSS up to three weeks in winter and spring. In JJA the positive skill is only observed for 2 weeks lead time and in SON for the first week only. Similar results are obtained for the Klöntal and the Thur catchment (not shown).

### 4.3 Streamflow prediction performance

#### 4.3.1 Prediction performance

The performance of the reforecasts is analysed for all available reforecast dates within the period 1994-2014. In Figure 4 the resulting scores (NSE, Bias, CRPSS and the spread-error-ratio) for all three catchments are presented indicating the skill of the prediction system against the reference simulation, i.e. the expected performance of the system for any date throughout the year.



For the Verzasca catchment, the skill in terms of the NSE from the ESP predictions suddenly drops after the initialization of the forecasts whereas the prediction system using meteorological reforecasts provides positive NSE up to 7 days. Pre-processing of precipitation generally even lowers the NSE indicating positive skill only for 5 lead days. Pre-processing of temperature does enhance the skill with positive NSE up to 13 days lead time. If both variables (precipitation and temperature) are pre-processed the positive effect of the temperature pre-processing on NSE is diminished by the negative effect of precipitation pre-processing.

The negative biases of the ESP approach indicate an underestimation of the streamflows for all lead times in the Verzasca catchment. Raw forecasts show even stronger underestimation. After pre-processing either precipitation or temperature this underestimation is lower and reach similar biases as the ESP predictions. If both parameters are pre-processed the biases are close to 0 or slightly positive indicating that the streamflows are neither under- nor overestimated.

The overall performance characterized by the CRPSS indicates positive skill for the ESP predictions up to 15 days lead time for the Verzasca, but the skill drops quickly after the initialization of the forecasts as in the NSE. Raw forecasts only show a positive CRPSS for the first 5 lead days. Pre-processing of precipitation increases the CRPSS at short lead times. Temperature pre-processing enhance the skill at early lead times and in addition elongates positive up to 15 days lead time. This is even more pronounced if both variables are pre-processed.

The spread error ratio of the ESP predictions is below 1 for all lead times indicating overconfidence. For the NWP Hydro chain the overconfidence is even higher for raw and temperature-only pre-processed (ppT) reforecasts. Pre-processed precipitation (ppP) reforecasts can partly correct the overconfidence of the streamflow reforecasts and if both variables are pre-processed (ppTP), the spread error ratio indicates reliable forecasts.

These skill signatures are similar in the other two catchments analyzed in this study (Figure 4), although not in an absolute sense. In the small, semi-glaciated Klöntal catchment the absolute skill generally is higher, and the skill of the predictions extends to longer lead times. In particular the raw and temperature-only pre-processed reforecasts (ppT) show positive skill in terms of the NSE throughout all lead times. In the Thur catchment the skill of the raw reforecasts outperforms the ESP predictions as well, but in contrast to both other catchments, the effect of pre-processing is negligibly small.

**4.3.2 Seasonal variations in skill**

The prediction skill of the different approaches does not only vary across catchments but as well across seasons. In Figure 5 the performance in the Verzasca catchment for the four seasons DJF, MAM, JJA and SON are shown. The general characteristics are similar as observed for the entire year, i.e. the ESP based predictions exhibit a sudden drop in the NSE after initialization and most benefits (positive skill to longer lead times) are obtained if temperature-only is pre-processed (ppT). In DJF and MAM this skillful horizon is extended by ppT, from 5 (3) days to up to 16 (9) days in MAM (DJF) and the bias is reduced.

The overall performance (CRPSS) for the ESP predictions is better than the reference climatology for all lead times in MAM and clearly outperforms the raw reforecasts and precipitation-only pre-processed reforecasts (ppP) in MAM and DJF. For the



temperature-only pre-processed reforecasts (ppT) and if both variables are pre-processed (ppTP), the predictions in these seasons (DJF and MAM) outperform the ESP forecasts for lead times up to 12 to 15 days and are equal for longer lead times. In JJA (and SON, not shown) the ESP predictions only exhibit a positive CRPSS up to 5 and 9 days, whereas the raw reforecasts (i.e. without any pre-processing) indicate positive skill up to lead times of 10 up to 15 days. However, in contrast to DJF and

MAM the influence of the pre-processing on the performance is negligible in JJA. Furthermore, in JJA the bias and the spread error ratio are only better if the pre-processing includes precipitation.

The seasonal variation in performance holds true as well for the other catchments, i.e. reforecasts initiated in winter and spring show highest benefits over the reference climatology. In contrast to the Verzasca catchment, the raw and precipitation only pre-processed reforecasts show higher streamflows than the reference climatology in MAM and the raw and temperature pre-

processed reforecast show a positive NSE up to 30 lead time days (Figure 6, left).

Less seasonal variation is observed in the Thur catchment although the general signatures are evident as well. In MAM (Figure 6, right) all methods perform better than the reference simulation (positive CRPSS) over the full forecast range. Generally worst performance is found for the ESP. The effect of the pre-preprocessing is limited and can mainly be noticed if precipitation is pre-processed, resulting in smaller biases and a Spread to error ratio closer to 1, indicating a reduction in overconfidence.

### 4.3.3. Reliability of the ensembles

An additional important forecast characteristic is the reliability of the predictions, which cannot directly be deduced from the metrics shown above. Therefore, the rank histogram for the full period (Full year) and MAM reforecasts in the Verzasca catchment for all model configurations is shown in Figure 7 to assess the reliability of the streamflow forecasts. As an example we focus on the rank histograms of the full period (Full year) and the MAM reforecasts, because of its representativeness for

the seasonality of performance. In MAM, both versions, raw and precipitation-only pre-processed reforecasts, show an underestimation of the flows (negative bias) indicating that most reforecast members tend to be lower than the corresponding observations. The strong negative bias is reduced if temperature-only pre-processed reforecasts (ppT) are used. But still a U-shape is evident in the histograms that indicates overconfidence and thus confirms the conclusions from the spread error ratio. If both temperature and precipitation is pre-processed (ppTP) the resulting rank histograms become more uniform, in particular

for longer lead times indicating a reduction of the overconfidence. But shorter lead times still exhibit some overconfidence. The rank histograms for the ESP predictions do provide more uniform rank histograms with a weak tendency of a negative bias. Although slight differences can be observed between different seasons, the main characteristics are similar for the full period.

The rank histograms for the Klöntal and Thur catchments (supplementary material) exhibit the same general behaviour

regarding the reliability, but improvements by pre-processing are less pronounced for the Thur catchment and the rank histograms still indicate overconfidence even if both variables are pre-processed.



### 4.3.4 Snow water equivalent verification

To generate skillful streamflow predictions in mountainous catchments, the snow in the catchment is a crucial variable. Therefore, the snow water equivalent (SWE) in the hydrological model is analyzed according to different elevation regions. For the verification we analyze the SWE at elevations above and below 1500 m a.s.l.. As in the verification of the streamflow reforecasts, the SWE is verified against the SWE of the reference simulation to replace the observations. For an evaluation of the SWE against real observation the reader is referred to Jörg-Hess et al. (2014).

In MAM, raw and precipitation-only pre-processed (ppP) reforecasts highly overestimate the SWE in areas above 1500 m a.s.l. indicated by the MAE and the bias in Figure 8. The predictive skill in terms of the CRPSS is lost after 9 days lead time. In contrast, the reforecasts in DJF show stronger overestimation in areas below 1500 m a.s.l. and a total loss of predictive skill after 15 days lead time in this area. Lowest biases and highest skill (in terms of the CRPSS) is evident for reforecasts with pre-processed temperature and precipitation (ppTP), followed by temperature-only pre-processed (ppT) both outperforming the ESP predictions.

For all versions of the meteorological reforecasts (raw and pre-processed) the resulting SWE reforecasts tend to be overconfident, with least overconfidence if precipitation-only is pre-processed according to the spread error-ratio. The ESP predictions exhibit less overconfidence for both seasons and regions and exhibit similar levels in terms of MAE and the bias and slightly less overall skill (CRPSS) compared to ppT and ppTP reforecast versions.

The rank histograms confirm the conclusion drawn above. Raw and precipitation-only pre-processed reforecasts (ppP) exhibit largest positive biases throughout all lead times. In case of temperature-only (ppT) and temperature and precipitation pre-processed reforecasts (ppTP) the rank histograms indicate overconfidence in the beginning which is reduced for longer lead times.

The SWE verification in the Klöntal catchment shows a similar behavior with negative biases and largest MAE for the raw and precipitation-only pre-preprocessed (ppP) reforecasts. In contrast to the Verzasca catchment the CRPSS stays positive for all versions and for all lead times. A similar behavior is observed in the Thur catchment with positive skill for all lead times but smaller negative biases (the corresponding figures for the Klöntal and the Thur catchment can be found in the supplementary material).

### Discussion

For a proper evaluation of the effect of pre-processing on the hydrological streamflow predictions the following discussion considers the verification against the reference simulation. The meteorological input reforecasts highly benefit from the pre-processing procedure applied. The skill found for the pre-processed temperature and precipitation reforecast is comparable to the skill found in Monhart et al. (2018). In contrast to the present analysis, our earlier study used a station wise post-processing of the raw forecasts using the same setup as in the present study. Similarly, different studies emphasize the benefit of pre-processing precipitation (Crochemore et al., 2016) and temperature forecasts (Lucatero et al., 2017) at catchments at various



spatial scales. The QM method used here is a popular pre-processing method for hydrometeorological ensemble forecasts (e.g. Kang et al., 2010; Lucatero et al., 2018; Verkade et al., 2013) but does not come without limitations. In particular Zhao et al. (2017) point out the inability of QM to provide fully reliable ensembles for post-processing GCM precipitation. However, an extensive discussion of the reliability issue of the pre-processed meteorological input data used in this study can be found in

Monhart et al. (2018). To summarize, it was found that QM indeed is able to provide reliable ensembles for lead times beyond 10 days but at shorter lead times the ensembles tend to be overconfident.

Our results show that subseasonal streamflow predictions in mountainous catchments can be skillful for the full 32 days lead time horizon in winter and spring. The traditional ESP approach clearly provides skillful predictions for all three catchments analyzed in this study, in terms of the overall skill (CRPSS) and the reliability. This is in agreement with the findings from

Arnal et al. (2018) comparing the skill the of an ESP and a seasonal forecasting system across many regions in Europe. They found that the ESP approach can be outperformed mainly in the first month in terms of the CRPSS. Nevertheless, if scores evaluating the mean characteristics are considered (NSE and bias) we observe worse performance than suggested by the CRPSS. This indicates that the ESP predictions can capture the future evolution of the streamflow in a probabilistic sense. Furthermore, the substantial decrease in skill within the first days suggest that the ESP predictions are not able to capture the

exact evolution but can rather be used to estimate the general behavior within the upcoming weeks. This is in agreement with the exponential decay in skill with increasing lead time found for ESP forecasts in UK catchments (Harrigan et al., 2017). If the NWP predictions are used to predict the streamflows, the skill can clearly be enhanced but in most cases only if the driving meteorological predictions are pre-processed. This indicates that the knowledge of the synoptic conditions plays an important role to enhance the skill of the streamflow predictions at early lead times and that biases in these driving predictions need to

be corrected prior to make these predictions useful.

The effect of pre-processing is even more pronounced for the SWE verification. In the NWP chain the SWE is highly overestimated if temperature is not pre-processed (raw and ppP). Hence, the hydrological model inherent downscaling of temperature using an adiabatic laps rate leads to low skill in terms of CRPSS at longer lead times. Although, temperature lapse rate corrections have been found to be important for reproducing streamflows simulation based on regional climate model

outputs in mountainous snow- and glacial dominated catchments (Butt and Bilal, 2011; Rahman et al., 2014) our study suggest that at least in a subseasonal forecasting context such corrections are not sufficient. Similarly, Tobin et al. (2011) have shown for flood forecasting framework constant lapse rate corrections even if seasonally-derived are unable to capture the dynamics of temperature changes during an event. At lower elevations this effect is even more crucial because the SWE is smaller and temperature biases accelerate melting processes in the model. If temperature pre-processing is included, these large errors can

be avoided, and the skill of the SWE predictions is substantially increased. This effect underlines the importance of pre-processing the subseasonal forecast in snow dominated catchments. The importance of the SWE initial conditions for subseasonal forecasts has been shown by Jörg-Hess et al., (2015). They conclude that a better representation of snow melt process by improved states of the snow storage can greatly improve the predictions of streamflow volumes. The influence of initial conditions of SWE on seasonal streamflow predictions in the US is shown by Wood et al. (2016) in an idealized





experiment. Furthermore, they stress that limited skill in seasonal meteorological can be amplified in streamflow prediction skill. Our study suggests that an additional pre-processing of the meteorological forecasts is necessary to maintain the benefit of the initial conditions and confirms the findings of amplified skill in the streamflow predictions if the forecasts exhibit some skill. Otherwise, low skill in performance of the (raw) forecasts and the loss in skill to predict the SWE over longer lead times directly translates into low skill of the streamflow forecasts.

The comparison of the performance in the three different catchments analyzed gives further insight in the predictability of streamflows in alpine catchments and strengthens the picture drawn above which shows the importance of the complex interactions between precipitation, temperature and SWE at the subseasonal forecast time scale. In the Thur catchment, the largest catchment in the analysis, which is mainly precipitation dominated, the difference in skill between the ESP predictions and the NWP based prediction is smallest. Similarly, the differences between raw and pre-processed experiments are marginal. Only in spring, if precipitation is pre-processed (ppP and ppTP) a small improvement in the CRPSS can be observed at early lead times coinciding with the lead times where pre-processed precipitations still exhibit skill. Hence the initial conditions of the model and skillful precipitation predictions at early lead times determine to a large degree the skill of the reforecasts and the negligible snow-covered area within the catchment does marginally affect the performance of the runoff predictions. In the Thur catchment two additional subcatchments in the upper Thur catchment (at the runoff stations Halden and Murg) have been verified to identify the influence of hilliness and catchment size on the forecast performance. Although the size of the Murg (212 km$^2$) and the Halden catchment (1750 km$^2$) varies significantly, the hilliness in both is comparable, while the lower part of the Thur catchment (station Andelfingen, 1696 km$^2$) is a typical lowland region. For all three stations the skill is very similar (supplementary material Figure S4) and pre-processing does not vary either. This suggest that neither hilliness nor catchment size does significantly influence the performance of the forecast. In the snow dominated and partially glaciated Klöntal catchment, the smallest catchment in this analysis, the high skill in terms of the NSE indicates a good performance of the ensemble mean but in terms of the overall skill (CRPSS) the NWP prediction are only skillful if temperature pre-processing is considered. This superiority in the mean is most likely the effect of melting processes. On the other hand, the Verzasca catchment is snow dominated in elevated regions, but rain dominated at lower elevations due to its large gradient of elevations. Therefore, snow-melt processes tend to occur more often and are more diverse due to the higher elevation gradient within the catchment. The NSE does indicate skill up to 13 days lead time for the temperature-only pre-processed forecast (ppT) but precipitation pre-processing even lowers the skill despite an increase in skill of the corresponding precipitation inputs. This counterintuitive behavior of lowering the skill (in terms of NSE) in the streamflow prediction despite the use of improved precipitation inputs underlines that for a profound assessment of the skill of ensemble forecasts, verification metrics focusing on mean flows can be insufficient and misleading.

The combination of pre-processed subseasonal meteorological prediction with hydrological simulations can outperform a traditional ESP approach in small to medium-sized alpine catchment. Especially in snow dominated and (semi-) glaciated catchments such a prediction chain brings large benefits in the forecast performance. But temperature (and precipitation) from the NWP model need to be pre-processed prior to be used in hydrological models to achieve better performance than an ESP




approach. In precipitation dominated catchment the pre-processing does only show a marginal improvement in skill, but the NWP chain clearly outperforms ESP predictions. Hence, such systems can be of interest for application when accurate and reliable runoff predictions are desired, especially in snow-dominated catchments. Furthermore, Frei et al., (2018) have shown that a general decrease of snow fall is expected in future climate change scenarios, while at higher elevation the signal shows

a slight increase in heavy snowfall events, due to a shift of climatological cold areas into a temperature interval which favor higher snowfall intensities in combination with a general increase in winter precipitation. Especially regarding future scenarios which expect an increase in hydro power production due to melt water in the period from October to April (Weingartner et al., 2013) such systems might become a valuable tool for optimizing hydropower production in mountainous areas. Future work should include statistical post-processing techniques (of the hydrological output) to correct the errors and biases of the

hydrological simulation.

The discussion above focused on the effect of pre-processing hydrometeorological prediction and therefore only the verification against the reference simulation was considered and hydrological model errors are thus excluded from the analysis. However, to estimate the real world performance the hydrological model errors need to be taken into account. To do so we verified the streamflows of the reference simulation of the PREVAH model against observations. The evaluation presented in

section 4.1 revealed the good performance of the hydrological model with NSE above 0.8 in most seasons. Largest difficulties are observed in seasons with low flow conditions (in DJF in the Verzasca, and in SON and DJF in the Klöntal catchment). This is particularly evident in the logarithmic version of the NSE, in which flood peaks are flattened to better asses the performance under low-flow conditions, the difficulties of the model remain. These deficits of the hydrological model need to be considered in the verification of the predictions and are the reason for verifying the reforecasts against the pseudo

observations from the reference simulation. Otherwise, if the predictions are verified against real streamflow observations the hydrological model deficiencies dominate the skill characteristics of the predictions and possibly impede the identification of the effect of pre-processing. To illustrate this, the verification was repeated with the real runoff observations. The skill in most seasons (MAM, JJA, SON) exhibit the same behavior with high skill at early lead times and decreasing skill at longer lead times. But during low flow condition (in DJF in the Verzasca catchment and in DJF and SON in the Klöntal catchment), strong

negative peaks in skill (in terms of the CRPSS) are present at short lead times, with increasing skill at longer lead times (supplementary material). The corresponding input data revealed that this behavior can be associated with predicted snow melt events that coincide with observed "no melt" events. During such events, all members of the experiments including pre-processed temperatures (ppT ad ppTP) overestimate the observed runoff peak. The comparison of the meteorological input data from the reforecast with data from an observational station shows an overestimation in both temperature and precipitation.

The temperature observations for this event were clearly below freezing, whereas the raw forecast data is close to 0° C, and the pre-processed temperature is positive. Hence, the melt-affected area is too large leading to an overestimation of the runoff contribution. In addition, overestimated precipitation will further contribute to runoff and less deposition of solid precipitation will occur. Similarly, the reference simulation does overestimate the runoff peak because of an overestimation of the temperature in the gridded data used to run the hydrological model. It is known that gridded observations do inherit errors due



to interpolation (Freudiger et al., 2016) and do have difficulties in resolving small-scale features, such as for example cold pools in alpine valleys (Frei, 2014). This can partly explain this behavior. Another potential explanation is an insufficient formulation of the discrimination between rain and snow in the hydrological model. In the version of PREVAH used for this study, the formulation follows a threshold based method in combination with a linear range as described in Zappa et al. (2003).

This linear transition range is set between the threshold values -1.5°C and +1.5°C set to +/-1.5°C as determined by the hydrological calibration. The threshold itself and the corresponding linear range highly depends on the hydroclimatic characteristics and thus can strongly vary in space (Liu et al., 2018). It has been shown that more sophisticated approaches using logistic regression for characterizing this range can provide better results (Frei, 2016). In principle such an approach could be included into the hydrological model, but such an implementation is out of scope of this analysis. Alternatively, it

has been shown that such errors of the hydrological model can be corrected by additionally post-processing the hydrological output using neuronal networks or logistic regressions (e.g.
Bogner et al., 2016; Sharma et al., 2018).

**Conclusion and outlook**

Recent advances in subseasonal meteorological ensemble models makes it feasible to develop hydrometeorological prediction

systems driven by such NWP forecasts. We developed an end-to-end hydrometeorological prediction system driven with reforecasts from the subseasonal prediction system from the ECMWF. A pre-processing procedure based on QM is used to bias correct and downscale the meteorological predictions prior to the hydrological model. The performance of the resulting streamflow forecasts is assessed for three small to medium size alpine catchments using various verification metrics to assess different attributes of the reforecasts. Our study demonstrates the potential of ensemble streamflow predictions in small

mountainous catchments. Moreover, the benefits of combining NWP predictions and hydrological models has been shown. The analysis indicates the need for pre-processing of the driving meteorological prediction especially in small snow-dominated catchments in alpine regions.

Decent skill of traditional ESP predictions compared to climatological reference can extend up to the entire 32 days. The NWP approach outperforms the EPS predictions in all catchments and most seasons in particular at short lead times up to about day

5. In snow dominated catchments, an additional pre-processing step of both, temperature and precipitation, is crucial to further enhance the skill and the reliability of the forecasts. While pre-processing precipitation-only is not sufficient to enhance the forecast performance, it is crucial in the combination with temperature preprocessing to improve the forecast reliability. Again, it is noted here that the verification is done against the reference simulation as replacement of real observations. Hence, the performance cannot directly be interpreted as the prediction performance in an absolute sense, because in our approach the

deficits of the hydrological model are not fully taken into account. However, the relative benefits of using NWP output as forcing for the hydrological simulations and the improvements after pre-processing is expected to hold true as well with real observations.



The benefits of the NWP approach and the pre-processing step is most pronounced in winter and spring when snow melt processes dominate. This demonstrates the importance of snow for the predictability of streamflows at the subseaonal timescale. Hence the deficits in the hydrological model with respect to snow related processes (in particular the distinction between solid and liquid precipitation) should receive further attention to enhance the forecast performance. Alternatively,

post-processing techniques applied to the streamflow forecasts can be applied to correct such hydrological model deficits. This would allow assessing the skill of the forecasts with respect to real observations and can potentially further increase the performance of the forecasts. Hence, if post-processing techniques are able to account for the deficits of the reference simulation, the combination of both pre- and post-processing could provide skillful lead streamflow predictions in snow and glacial dominated catchment in mountainous terrain at the subseasonal forecast horizon.

Furthermore, technical improvements of the NWP models related to the ensemble size, frequency of issuing reforecasts and improvements in the representation of physical processes, can be expected to have a positive effect on the resulting streamflow performance. In our setup we use a rather simple statistical bias correction technique to pre-process the hydrometetroloigcal prediction. More sophisticated pre-processing techniques could be applied to analyze their capability to enhance the streamflow performance. Since ensemble hydrometeorological predictions are of interest for specific applications the forecasts should

further be analyzed according to their economic value. For example to optimize the revenues of existing hydropower plants in alpine regions or for early better preparedness of hydrological droughts.

Data availability:

ECMWF forecast data are accessible through the MARS archive: http:/ /apps.ecmwf.int/archive-catalogue/, observation

records used in this study are available from the following website: https://gate.meteoswiss.ch/idaweb/more.do (SwissMetNet). Streamflow series and catchment boundaries are provided by the Swiss Federal Office for the Environment (FOEN).

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



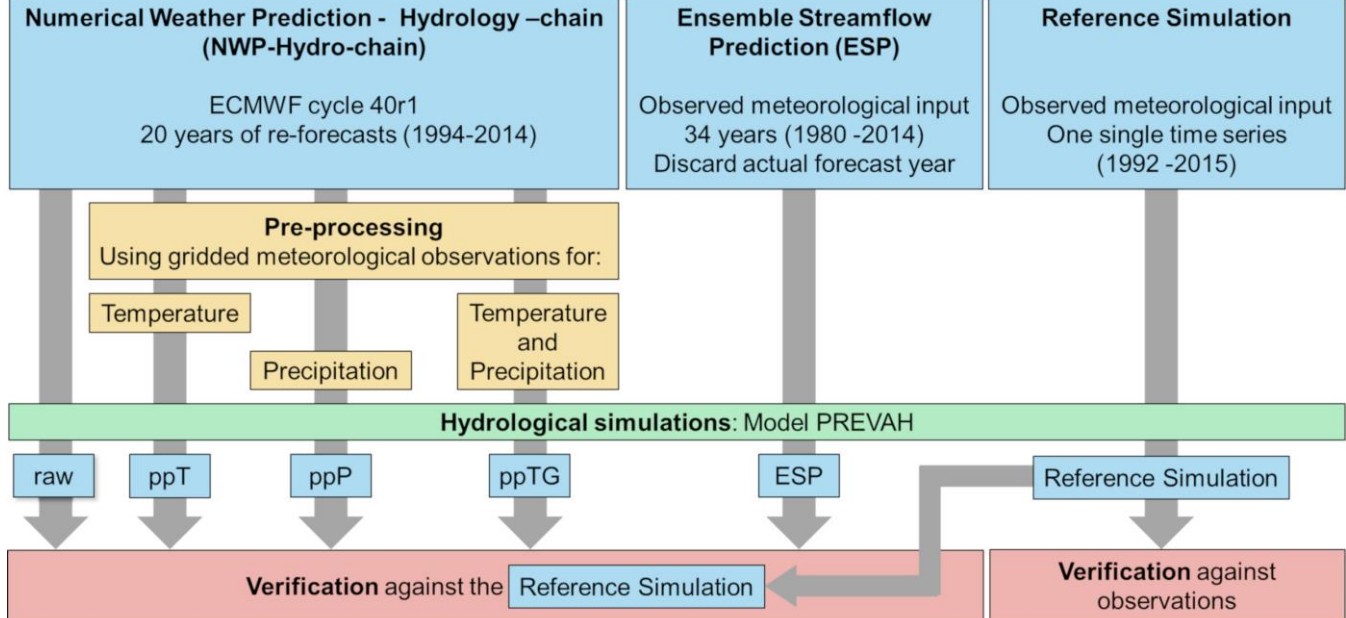

Figure 1: Conceptual design of the NWP-Hydro-chain with and without pre-processing (left) and the ESP chain (right).



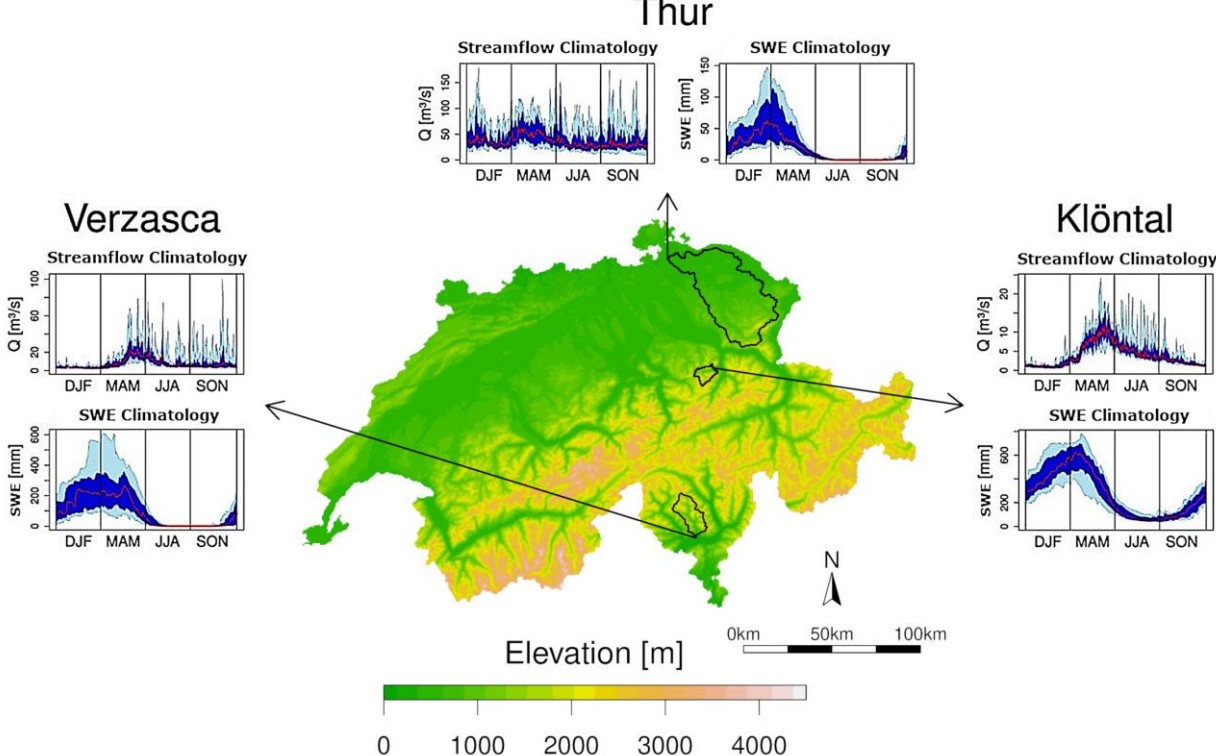

**Figure 2: Location and extent of the three selected catchments and their corresponding streamflow and SWE climatology. The Verzasca catchment in the south, Klöntal catchment in the center and the Thur catchment in the north east.**





|  |  | Verzasca | Klöntal | Thur |
|---|---|---|---|---|
| **Catchment area [km2]** |  | 185 | 83 | 1696 |
| **altitude range [m]** | maximum elevation | 2864 | 2883 | 2505 |
|  | average elevation | 1651 | *NA* | 770 |
|  | minimum elevation | 490 | 847 | 356 |
| **dominant hydroclimatic regime** |  | snow | snow and glacial | precipitation |

Table 1: Overview of the catchment characteristics for the three catchments analysed.

|  |  | Verzasca |  | Klöntal |  | Thur |  |
|---|---|---|---|---|---|---|---|
| **NSE** | **FullYear** | **0.85** |  | **0.84** |  | **0.84** |  |
|  | DJF | 0.38 |  | 0.39 |  | 0.83 |  |
|  | MAM | 0.88 |  | 0.82 |  | 0.85 |  |
|  | JJA | 0.82 |  | 0.75 |  | 0.79 |  |
|  | SON | 0.84 |  | 0.75 |  | 0.86 |  |
| **NSE log** | **FullYear** | **0.87** |  | **0.68** |  | **0.87** |  |
|  | DJF | 0.44 |  | -0.14 |  | 0.82 |  |
|  | MAM | 0.90 |  | 0.84 |  | 0.87 |  |
|  | JJA | 0.86 |  | 0.78 |  | 0.85 |  |
|  | SON | 0.89 |  | 0.43 |  | 0.88 |  |
| **MAE [m³/s] ([%])** | **FullYear** | **2.84** | **(25.9)** | **1.19** | **(26.6)** | **10.54** | **(22.5)** |
|  | DJF | 1.35 | (36.1) | 0.73 | (50.7) | 10.89 | (25.2) |
|  | MAM | 3.16 | (18.7) | 1.20 | (19.1) | 11.53 | (16.2) |
|  | JJA | 3.40 | (33.9) | 1.63 | (26.7) | 11.30 | (28.7) |
|  | SON | 3.45 | (23.1) | 1.21 | (39.6) | 8.50 | (23.9) |
| **Bias [m3/s] ([%])** | **FullYear** | **0.28** | **(2.5)** | **0.26** | **(5.9)** | **-0.42** | **(-0.9)** |
|  | DJF | 1.07 | (28.7) | 0.39 | (27.2) | 2.49 | (5.8) |
|  | MAM | -0.06 | (-0.4) | -0.01 | (-0.2) | -1.47 | (-2.1) |
|  | JJA | 0.23 | (2.3) | 0.04 | (0.6) | -4.41 | (-11.2) |
|  | SON | -0.11 | (-0.7) | 0.66 | (5.9) | 1.74 | (4.9) |

Table 2: Verification of the reference simulation with corresponding observations for the Verzasca catchment, the Klöntal catchment and the Thur catchment. In each catchment the Nash-Sutcliffe coefficient (NSE), the Nash-Sutcliffe coefficient using logarithmic input values (NSE log), the bias and the mean absolute error are shown. The verification is done for the full simulation period 5 (FullYear: 1994-2014) and the individual seasons within the full period (DJF, MAM, JJA, SON). A perfect simulation would have NSE=1, and positive values of NSE indicated better skill than the reference climatology



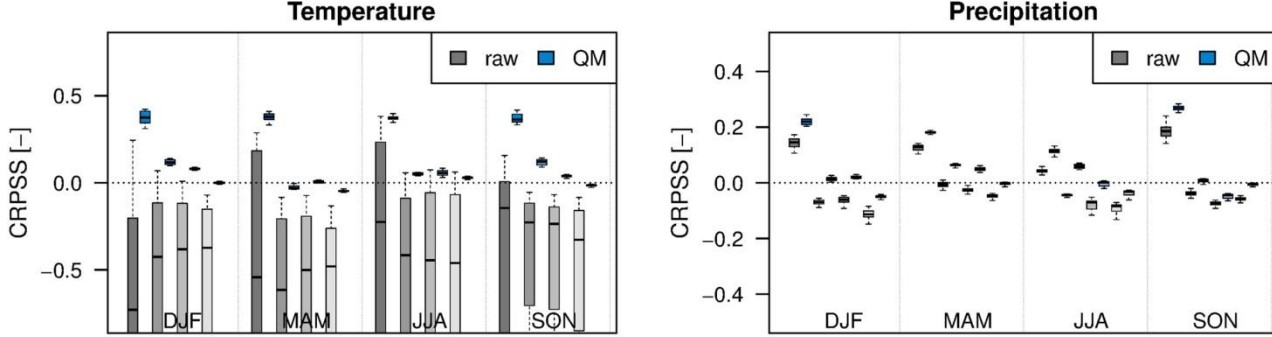

**Figure 3: Overview of the CRPSS for weekly mean temperatures (left) and weekly precipitation sums (right) in 1994-2014 reforecasts grouped by season (DJF, MAM, JJA and SON) for all grid points within the Verzasca catchment. The shading of the boxes denotes lead time, whereas week 1 corresponds to day 5-11, week 2 to days 12-18 and so on. Grey shading for raw forecast and blue shading for pre-processed reforecasts. An individual box shows the distribution of the CRPSS for all grid points within the catchment averaged over all 13 reforecast initialization dates within one season. The boxes depict the interquartile range, the mean is indicated by the horizontal line and the whiskers span the length of 1.5 x standard deviation of the data. A perfect forecast has CRPSS=1, and positive values indicate better skill than the reference climatology**





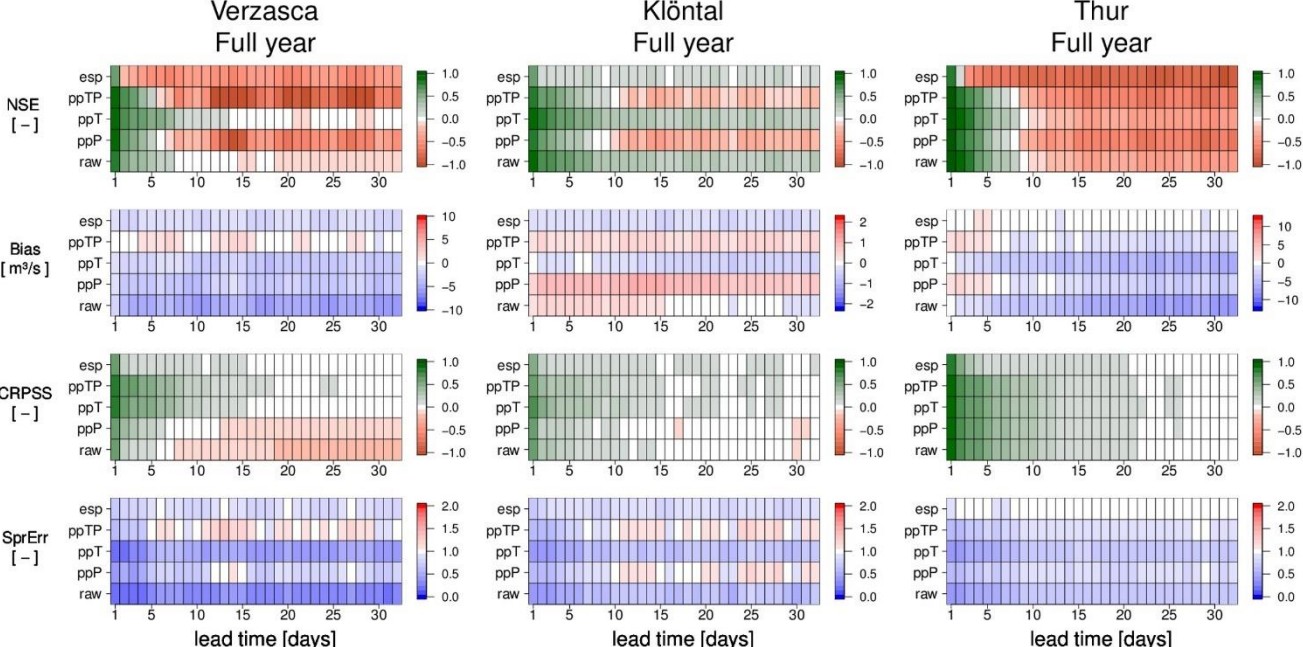

**Figure 4: Verification of the streamflow forecasts for the Verzasca (left), the Klöntal (middle) and the Thur catchment (right) considering all forecasts within the reforecast period. The Nash-Sutcliffe coefficient (NSE) in the upper most panel, the mean bias (Bias) in second, the CRPSS in the third, and the Spread-Error relationship in the lowest panel. For each score the 5 different setups are shown. The first row corresponds to the ESP approach (esp), the second row to the reforecasts using both pre-processed temperature and precipitation (ppTP), the third row to the pre-processed temperature-only (ppT), fourth row to the pre-processed precipitation-only (ppP) and the fifth row to the reforecasts using the raw meteorological input reforecasts (raw).**





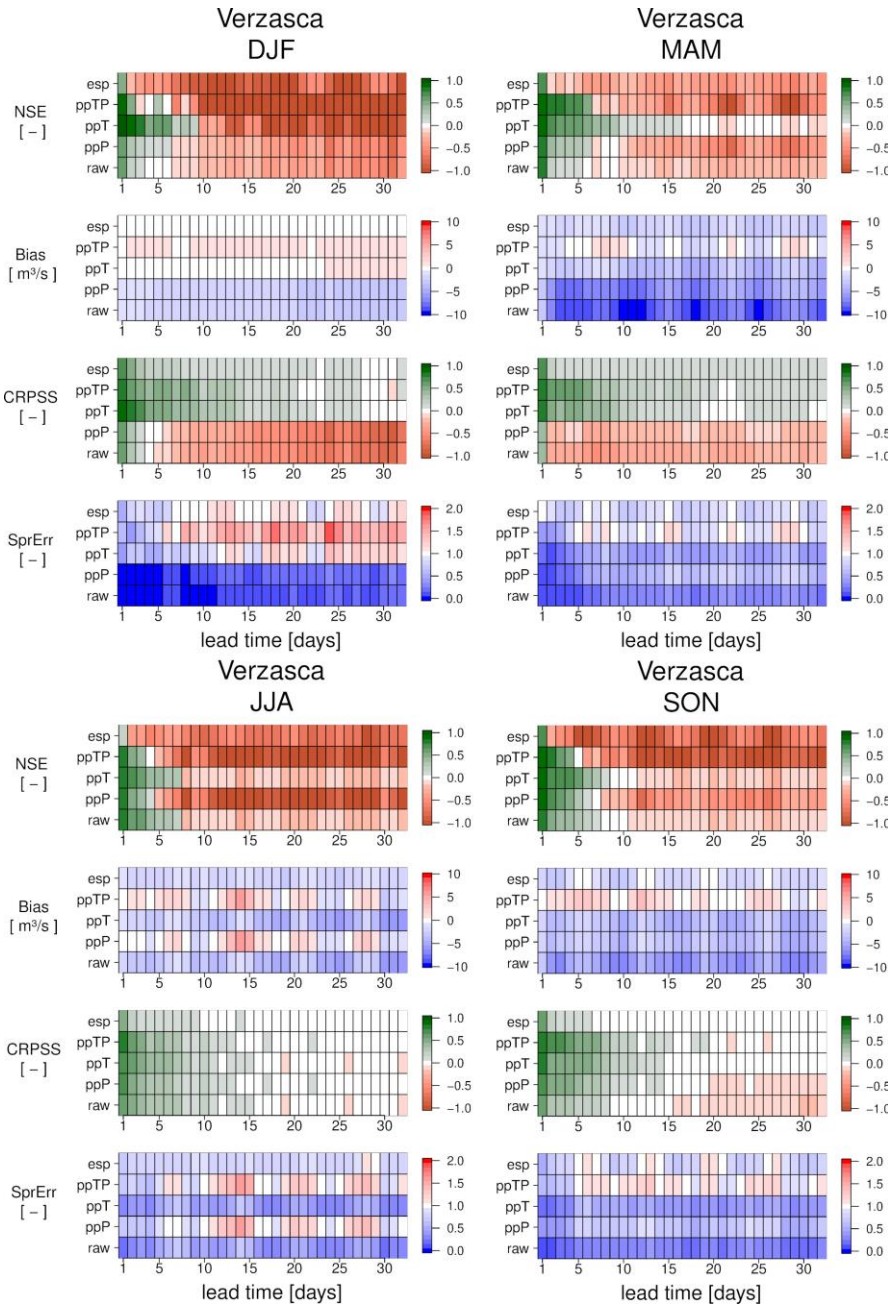

**Figure 5: Same as Figure 4 but seasonally aggregated (DJF, MAM, JJA and SON) for the Verzasca catchment.**





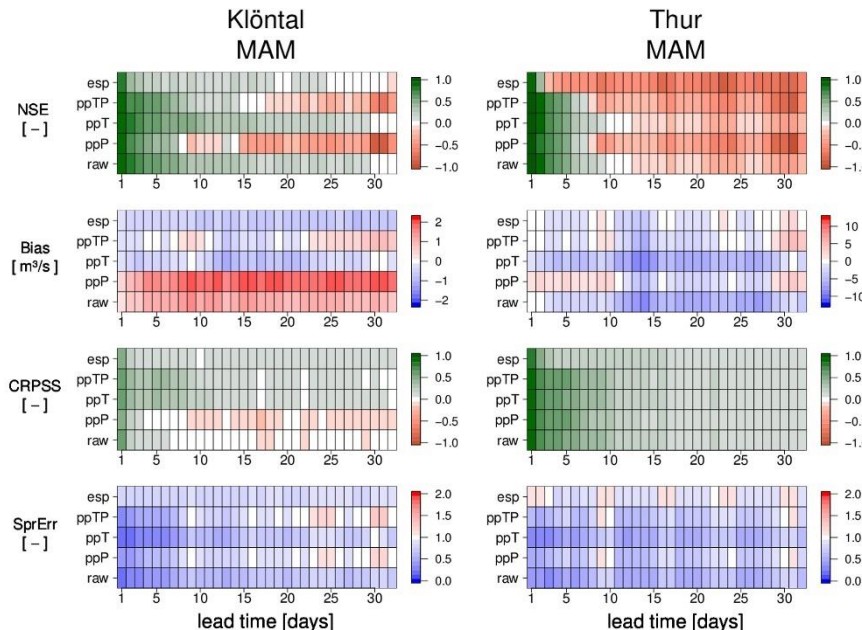

**Figure 6: Same as Figure 4 but for the Klöntal and the Thur catchment in MAM.**





**Figure 7: Rank histograms for all four configurations (raw, ppP, ppT, ppTP) and the ESP predictions for the full analysis period (Full year) and MAM in the Verzasca catchment. The basic principle in the rank histograms the assumption that the ensemble members determine bins in which the corresponding observation can be ranked. For a reliable forecast, the observations are equally distributed across all different bins resulting in a uniform shape of the rank histograms. If for example the frequency in the lowest (rank 0) and the highest bin (rank 6) is much higher, the observation tends to be more frequently either higher than all ensemble members or lower than all ensemble members but less often in between the ensemble members. This specific U-shape indicates that the forecast spread is too narrow and thus the forecasts generally overconfident. In contrast, if the observations tend to be more often in between the ensemble members (e.g. rank 2 and 3) the rank histogram exhibit a convex shape and thus the forecast spread is too large indicating overdispersive forecasts.**

**Figure 8: Same as in Figure 5 but for SWE in DJF and MAM in the Verzasca catchment.**