# Peer review of "Subseasonal hydrometeorological ensemble predictions in small- and medium-size mountainous catchments: Benefits of the NWP approach"

_Hydrology and Earth System Sciences, 2018_

## Referee Comment (RC1) · Anonymous Referee #1 · 19 Oct 2018

Review on the paper by Samuel Monhart et al. Subseasonal hydrometeorological ensemble predictions in small- and medium-size mountainous catchments: Benefits of the NWP approach Presented for the review is a paper dedicated to the evaluation of subseasonal streamflow forecasts performance in three mountainous catchments in Switzerland produced by the two approaches. Both approaches involve a regional hydrological process-based model PREVAH to account for the initial conditions in the catchment under consideration and the main concern of the paper is concentrated in the model driving for the forecast lead-times. The first approach is the Ensem-

ble Streamflow Prediction (ESP) framework as described by Day (1985), which uses the historical weather data to force the model for the forecast lead-time resulting in the ensemble of the streamflow hydrographs. The second approach is the Numerical Weather Prediction (NWP) framework that involves a meteorological large-domain model-based 5-member forecasts by the ECMWF IFS to force the hydrological model for the lead-time period. The authors use both the raw and bias-corrected NWP forecasts in terms of meteorological and hydrological forecasting skill. The performance of the forecasting approaches is evaluated both for deterministic and probabilistic properties, e.g. the average characteristics are benchmarked by the MAE, NSE and NSElog criteria and the ensemble spread is evaluated by CRPSS metric, as well as the forecast spread to error ratio; the reliability of the forecasts is further examined by constructing the rank histograms. The overall importance of the study is crucial beyond doubt, as is very well described in the Introduction section – the ensemble forecasting methodology is now employed in many forecasting centers around the globe, yet the mentioned improvement in the NWP systems that hydrological prediction systems may benefit from is achieved mainly in Europe and North America, where the outstanding effort to it is applied. The case study catchments choice matches the research aims very well, as very diverse streamflow generation conditions are within the scope of the study – snowmelt-driven and fast-responding catchments are considered, which are an effort in constructing a well-performing streamflow model, as well as reliable subseasonal forecast, especially for summer and fall rainy periods with short hydrological system memory. Still, the authors show good model evaluation metrics. The main findings in the paper are in different effects of NWP bias-correction on the forecast performance, which vary in terms of variable, space and time, e.g. pre-processing of the input forcing is evaluated for temperature and precipitation apart and combined, and discussed for the three catchments over several seasons. The forecasts performance is evaluated not only for the streamflow but for the snow water equivalent in the catchments, as well, which is very crucial for the understanding of the predictability of snowmelt runoff. The results show the NWP pre-processed temperature forecasts outperforming

the ESP forecasts, which is a crucial finding, as well. There are a few concerns that I would appreciate the authors to enhance in the paper. First is the statement in section 3.2 concerning the minor importance of such variables as relative humidity etc. on the model performance on such timescales. Hence, the authors state that the relative humidity values were taken from the forecasts without any pre-processing. I would not agree with the authors on the minor importance of the relative humidity, as the evaporation rate is highly dependent on the relative humidity, especially within the processes of evaporation from snow. Given that the temperature forecasts are pre-processed, the close relation of the relative humidity and temperature may influence the forecast performance. However, the assessment of this was beyond the scope of the study. In the SWE forecasts verification section 4.3.4 the forecasts are verified against the reference model run instead of the actual observation, yet I would appreciate if the reference model performance could be discussed at least within a few sentences. Another consideration is that the methodology of the SWE assessment should be placed in the corresponding subsection within section 2. Minor technical note: p. 13 l. 1 - "...seasonal meteorological..." must be followed with a noun, which is missing

My overall perception of the paper is that it presents an outstanding scientific effort, which is of critical importance to the modern hydrological forecasting systems research. The motivation is well described, the methods are concise and well referenced, the results are well documented and discussed and a number of very crucial statements on the topic are made. I would recommend minor revisions before the paper can be published.

Please also note the supplement to this comment:
https://www.hydrol-earth-syst-sci-discuss.net/hess-2018-458/hess-2018-458-RC1-supplement.pdf

---

## Referee Comment (RC2) · Anonymous Referee #2 · 23 Oct 2018

General comments This is an interesting and thorough assessment of an ensemble streamflow forecasting system in snow-affected mountainous catchments. The system pairs NWP forecasts with a distributed hydrological model that includes detailed accounting of cryogenic processes. The system is technically advanced and in my opinion of high interest to the readership of HESS. The study is well conceived and very clearly written. A particular strength of the study is the extensive and thorough verification of the forecasts, encompassing multiple appropriate measures of performance that are described and discussed in clear and interesting ways. In general, the

authors' conclusions are strongly supported by their analyses. I have one quibble with the use of QM as a means for downscaling meteorological forecasts, which amounts to a minor revision. Other than this, I have no hesitation in recommending this study for publication.

Major comments It appears that the authors use QM to downscale NWP predictions from a coarse grid (∼30/60 k) to a fine grid (∼2 k). It's well established that QM is not theoretically ideal for this practice, because of so-called variance inflation (Maraun 2013). The authors appear to be aware of this, as they discuss this issue in an accompanying paper (Monhart et al. 2018). However, it is more salient in this paper, because of the hydrological modelling that is carried out. Variance inflation is only an issue when quantile mapped/downscaled meteorological forecasts are spatially re-aggregated, which is exactly what the hydrological model does. So it will not show up in the analyses carried out by Monhart et al. 2018 (where variables are not re-aggregated), but it could well be an issue in this study. In addition, and as Maraun shows, the variance inflation problem is only strongly evident for extremes. Extremes are not the focus of the analysis carried out in this paper, which is fine. But this means it's hard to tell if variance inflation is present in streamflow forecasts. As the forecasting system could be used for flood prediction, this may be a serious issue. Accordingly, I recommend two changes to the manuscript:

1) The authors should clearly describe how they bridge the gap in spatial resolution from a ∼30/60 k horizontal grid (NWP forecasts) to a ∼2 k horizontal grid (observations). And if, as I've assumed, they use QM for this purpose: 2) The authors should briefly acknowledge the issue of variance inflation in the discussion, including a discussion of possible implications for their system (perhaps alongside recommendations for dealing with these implications).

I also encourage the authors to consider measuring the impact of variance inflation on their system in future work.

Specific comments Page 2 L7-9 "For a given target day of a reforecast the correction is derived from the distribution of all the reforecasts within a three weeks window around the same lead day and the corresponding observations, hence the correction depends both on the lead time and on the period of the years". Is the QM cross-validated in some way? How are zero values in precipitation handled in the QM?

Page 5 Section 2.4 For each score used, please note the range of values taken and the orientation of the score (e.g. -infinity to 1 for NSE, with 1 being perfect) This allows easy intepretation of, e.g., Figure 4.

L16 "we use the spread to error ratio (SprErr) as an indicator for the forecast reliability" Please briefly describe how this is calculated.

Page 8 L18 Figure 3. It's very difficult to see the different colours in this figure, especially in the right hand panel - i.e., it's not possible to distinguish QM from raw. Please replot so this is clearer (e.g., with different colours/box outlines, and/or perhaps restrict the vertical axis in the rh panel to [-0.2 0.3]).

L22-23 "After bias correction the skill is higher with positive CRPSS up to three weeks in winter and spring." From the figure, precip skill looks to be negligible in DJF after week 1. Skill scores will of course be a little noisy (in time, as well as in space; the authors have only considere spatial variation), so I don't think the authors should describe forecasts as 'skillful' if they have CRPSS values only very slightly above zero.

L28 NSE, Bias - I assume these are calculated on the mean of the ensemble? Please state this in Section 2.4.

Page 9 L7-8 "The negative biases of the ESP approach indicate an underestimation of the streamflows for all lead times in the Verzasca catchment." It's not clear to me why ESP predictions would be biased. ESP forcings, by construction, are unbiased. As the bias in predictions is calculated against model climatology, there should be no bias, as occurs in the Thur. Please briefly explain what is going on here.

L15 "The spread error ratio of the ESP predictions is below 1 for all lead times indicating overconfidence." There are two issues here. First, as I already alluded to, it would be beneficial to readers unfamiliar with the spread-error ratio to offer a brief explanation of the range of values it can take, and which direction indicates over/under confidence in Section 2.4. Second, I can't understand why the ESP forecasts are not reliable. ESP forcings are by construction reliable, so the spread-error ratio for the ESP forecasts should be close to 1. But this is not so in the two smaller catchments. Why is this?

Page 10 L26-27 "The rank histograms for the ESP predictions do provide more uniform rank histograms with a weak tendency of a negative bias." Again, I would be interested in a brief explanation of this bias in ESP forecasts.

Page 12 L15 "QM indeed is able to provide reliable ensembles" To me this sounds as though QM is responsible for the reliable ensembles, and this isn't really correct. QM can improve reliability to the extent that it is negatively impacted by bias (including conditional bias - i.e., biases at different points in the marginal distribution). You have shown (like Zhao et al.) that QM does not correct for overconfidence in short lead-time forecasts. This is because the underlying forecasts are overconfident - QM can't correct this (by construction). At longer lead-times, the QM forecasts are reliable because the spread in the underlying forecasts is appropriate (notwithstanding conditional biases). The same goes for coherence (discussed in the accompanying Monhart et al. 2018 JGR paper evaluating the NWP forecasts) - QM is not capable of correcting negative skill wrt to climatology in forecasts, other than that due to bias (again, by construction). The forecasts presented in this study are coherent because the underlying (raw) forecasts appear to be neutrally skillful at long time scales. In other words, it is the combination of NWP model and the QM that create the reliable and coherent forecasts shown in this study, not just QM. I think it would be better to reword what's written here to reflect this fact.

P14 L9-10 "to correct the errors and biases of the hydrological simulation" and, presumably, to account for additional uncertainty induced by the hydrological model in the

ensemble?

P14 "To do so we verified the streamflows of the reference simulation of the PRE-VAH model against observations." One thing not discussed here is reliability. I assume when assessed against observations, the ensembles are highly overconfident because uncertainty in the hydrological model is not included in the ensemble (see, e.g., Bennett et al. 2014). This is especially true at very short lead times (perhaps <3 days), when hydrological model uncertainty may be the dominant source of uncertainty in the forecasts. This may be worth mentioning.

Typos/Grammar Page 1 L14 "Prior of" should be "Prior to" or more simply "Before"

Page 2 L11 "both," delete comma

Page 5 L2 "comparison of to" delete "of" L28 "year" should be "years"

Page 7 L3 "station" should be "stations"

Page 8 L9 "evaluate of operational" delete "of"

Page 9 L9 "and reach" should be "and reaches" L11 "enhance" should be "enhances" L11 "elongates positive up" I think "skill" is missing here - i.e. "elongates positive skill up" L27 "are shown" should be "is shown"

Page 13 L1 "in seasonal meteorological can" I think this should be "in seasonal meteorological forecasts can"

Page 14 L11 "prediction" should be "predictions" L23 "exhibit" should be "exhibits"

Page 15 L14 "enhance" should be "enhances" L24 "EPS" should be "ESP" L25 "both, temperature" delete comma

Page 27 L5 "whereas" should be "where"

References Bennett JC, Robertson DE, Shrestha DL, Wang QJ, Enever D, Hapuarachchi P, Tuteja NK. 2014. A system for continuous hydrological ensemble forecasting (SCHEF) to lead times of 9 days. Journal of Hydrology 519: 2832-2846. DOI: 10.1016/j.jhydrol.2014.08.010.

Monhart, S., Spirig, C., Bhend, J., Bogner, K., Schär, C. and Liniger, M. A.: Skill of Sub-seasonal Forecasts in Europe: Effect of Bias Correction and Downscaling using Surface Observations, J. Geophys. Res. Atmos., 1–18, doi:10.1029/2017JD027923, 2018.

Zhao, T., Bennett, J. C., Wang, Q. J., Schepen, A., Wood, A. W., Robertson, D. E. and Ramos, M. H.: How suitable is quantile mapping for postprocessing GCM precipitation forecasts?, J. Clim., 30(9), 3185–3196, doi:10.1175/JCLI-D-16-0652.1, 2017.

---

## Referee Comment (RC3) · Anonymous Referee #3 · 26 Oct 2018

General comments. The main motivation of this study is to fill the gap in small scale researches in determination of the propagation positive skill extent in meteorological prediction models further into the streamflow forecasts. To address this problem, a traditional ESP approach was compared with prediction driven by ECMWF subseasonal ensemble system in three alpine catchments with varying hydroclimatic conditions. To emphasis the effect of applying pre-processing (QM-based) of NWP output, prediction verification was done against the reference simulation (pseudo observations). Thus hydrological model errors were excluded from the analysis. Summary. There was indepth discussion on hydroclimatic variability and predictability, the role of forcing and model parameters' uncertainty. The verification metrics used were relevant and applied in a logical manner. The results well supported the conclusions. Some sections recommend recompiling for better logically organized and easy follow.

I recommend publishing the manuscript but encourage the authors:

1) Give a more justification on choice of these watersheds for sub-seasonal forecasting. Initially, it can be supposed that the study is a part of a large numerical experiment where the results were confirmed only for the three arbitrary watersheds. 2) Add meteorological observations network on fig. 2 and give some comments explaining the good modeling quality (tab. 1) when using the grid product obtained at a low observations network density. 3) Give a number of predictions made for evaluation. 4) Comment on how the processing of only temperature and precipitation affects and propagate through the hydrological simulation. 5) Specify if the ESP method can outperform the NWP if ensemble takes not all but only individual years guided by a certain criterion for the similarity of the initial conditions. 6) Recompile the sections 2 and 3 referred to each other to make them more consistent.

Specific comments.

P.13, L5. The upper (reads like nested) Thur subcatchment Halden (1750 km2) is little bit bigger then Thur watershed itself (1696 km2).

---

## Author Comment (AC1) · 16 Nov 2018

Thanks a lot for this extremely motivating review for our article. We highly appreciate your comments and will account for your suggestion in the revised version of the manuscript and we will provide detailed replies regarding your specific comments. We agree with your comments on the importance of other variables such as relative humidity and sunshine duration and will re-write it mentioning the importance of these variables for snowmelt. But a conclusive analysis of pre-processing relative humidity (and other variables) will be beyond the scope of this study. Regarding your comments

on the SWE verification we will include a brief discussion of the findings presented in Jörg-Hess et al. (2015) where they found good agreement between the predicted SWE and the observed SWE maps for the entire 32 day forecast horizon and will add a paragraph on the verification of the SWE in the methodology section 2 like the one about the verification of the streamflow.

Jörg-Hess, S., Griessinger, N., & Zappa, M. (2015). Probabilistic forecasts of snow water equivalent and runoff in mountainous areas. Journal of Hydrometeorology, 16(5), 2169-2186. https://doi.org/10.1175/JHM-D-14-0193.1

————————————————

---

## Author Comment (AC2) · 16 Nov 2018

Thank you for your comments and your valuable input. We will account for your suggestions in the revised version of the manuscript. Regarding the selection of the watersheds we want to stress that this is not an arbitrary choice but rather a compromise between the intendent application of our results (within hydropower optimization in the Alpine region) and to adequately meet the requirements of a scientific analysis. Hence, the Klöntal and the Verzasca catchment both are selected because of existing hydropower installation in these watersheds and the Thur catchment was chosen as a

representative catchment with different hydroclimatic characteristics and because the catchment is often considered in hydrological research in Switzerland. We will further highlight this in the revised version of the manuscript. Furthermore, we will give an explanation for the good modelling quality using the gridded observations and provide more information of the number of forecasts used (1040 forecasts with 5 members each are used, corresponding to 1 forecast a week for the full 20-year reforecast horizon). In addition, we will make some changes in section 2 and 3 to improve the distinction between these sections. We can add additional comments on how the pre-processing of only temperature and precipitation affects and propagates through the hydrological simulation by extending the existing discussion. Furthermore, we will add a discussion about the potential enhancement of the performance of the ESP predictions based on a selection of years where the initial conditions show high similarity with present conditions and put it in relation to existing literature as for example Crochemore et al. (2017) who showed that seasonal forecasts can benefit by conditioning climatology. However, a thorough assessment of conditioning the ESP predictions is beyond the scope of the present study but could be considered for further studies within this area. We will provide more extensive replies to your comments during the upcoming revision process.

Crochemore, L., Ramos, M.-H., Pappenberger, F., and Perrin, C.: Seasonal streamflow forecasting by conditioning climatology with precipitation indices, Hydrol. Earth Syst. Sci., 21, 1573-1591, https://doi.org/10.5194/hess-21-1573-2017, 2017.

---

## Author Comment (AC3) · 16 Nov 2018

Thank you for your valuable comments. Indeed, QM is not a universal method to bias correct and downscale meteorological predictions and does have its limitation (especially regarding the variance inflation as you stated in your comments). Hence, we agree with your critical points raised concerning the QM methodology and intend to adapt the manuscript accordingly by extending the discussions and point out the difficulties and limitations of the chosen method and providing suggestions for alternative approaches for the readers. Concerning the two recommended changes we plan to

include it in the following way. First, as you propose, we extend the description about how we bridge the gap between the resolution of the meteorological forecasts (with 50km spatial resolution) and the gridded observations (with 2km spatial resolution). In short, as you assumed correctly, we use QM in a cross-calibration framework for this purpose. Hence, we use the same cross-calibration framework as proposed in the paper by Monhart et al. (2018) but the point observations are replaced with the gridded observation data. However, we perform a bilinear interpolation to the surrounding grid data of the coarse forecast (for both the stations and the 2km grid) before applying the QM approach. Second, the topic of variance inflation is indeed important and, as you suggested, it might be of greater importance in this study when the forecasts are aggregated again within the hydrological system, compared to the study by Monhart et al. 2018 where single locations are used for the bias-correction and verification. As mentioned above, we first perform a bilinear interpolation from the coarse resolution forecast grid onto the 2 km observational grid, prior to the bias correction and downscaling in the leave-one-year out cross-calibration framework using QM. Hence, some spatial variability is induced and each of the 2km grid point does provide a slightly different information to be downscaled. This might reduce the spatial effect of variance inflation compared to a bias correction (and downscaling) using a nearest neighbour interpolation technique. However, we plan to have a closer look at the variance inflation within our prediction setup and will extend the discussion in the revised version of the manuscript to point out potential effects on the results.

In the following, we give short answers to your general comments and how we plan to include your points raised in the revised version of the manuscript:

Page 4 L7-9 As mentioned above, we do use a cross-calibration framework to bias correct and downscale. This will be included in the revised manuscript.

Zero values are not handled in a specific way. The reason is, that we apply a multiplicative correction where zero values will not cause an issue. With a multiplicative correction, QM does not artificially produce rain, i.e. in case zero precipitation is fore-

casted the corrected precipitation still provides zero precipitation. In general, weather prediction models exhibit a drizzle effect due to their large grid sizes, meaning that raw model forecast predict too much rain compared to the observation (in case of very low precipitation rates). Hence, zero precipitation values do not need special treatment during the pre-processing. We missed to include this information in the present form of the manuscript and will add this in the revised version.

Page 5 Section 2.4

For each score we will include the range of values to make the figures easier to interpret.

L16

The spread to error ratio is defined as the ratio between the variance of the forecast ensemble (forecast spread) and the mean squared error (MSE) of the forecast ensemble (forecast error). We will explicitly mention how we calculated the spread to error ratio ratio in the revised version of the manuscript.

Page 8 L18 Figure 3

We will increase the readability of the figures according to your suggestions.

L22-23

We agree that the formulation of this sentence is too optimistic and will adapted the sentence by only highlighting positive skill up to three weeks in spring.

L28

As you assume correctly we calculate the NSE based on the mean of the ensemble and the bias corresponds to the mean ensemble bias. We will highlight this accordingly in the method section.

Page 9 L7-8

In general, we agree that the ESP predictions should be reliable by construction as it is found for the Thur catchment. The reduced reliability as well as the bias could be a result of the combination of the two following characteristics. First, the climatology is represented by the average conditions within 20 years, based on the reference simulation. If, for a specific forecast instance, the initial conditions strongly deviate from the climatological estimate, the resulting ESP predictions tend to show a bias especially at shorter lead times until the influence of the initial conditions diminishes. This effect is expected to be more pronounced in snow dominated and fast reacting catchments (as the Verzasca and the Klöntal catchment), whereas in the larger, rain dominated catchments this effect will have less influence on the results. Second, the climatological reference period (1994-2014) and the period of the meteorological observations of the ESP predictions (1980-2014) do not exactly coincide with each other. The meteorological observations used for the ESP covers a longer period and may exhibit a trend in temperature that, in addition, is stronger pronounced between 1980 and 1990. This could influence our analysis and would be an argument to repeat the ESP prediction with using the same period as for the climatology. But on the other hand, to ensure an ensemble size that is large enough, we decided to use the full available period (1980-2014) to generate the ESP ensembles. However, we plan to discuss this in the revised manuscript and put it in relation to existing literature.

L15 As mentioned above, we will specifically include the range of the scores.

Page 12 L15

We will rephrase the sentence accordingly to make a clearer statement that the reliability in the corrected forecasts is a result of the combination of the NWP model and QM, and thus cannot be realized by QM alone.

P14 L9-10 We will rephrase this sentence to account as well for the uncertainty induced by the hydrological forecast model. Furthermore, this will be accounted for when the hydrological forecasts are in addition post-processed, which is intended in future work.

Your input to discuss the reliability of the ensembles verified against observations will be included in the revised manuscript. As you expected, if the ensembles of the reforecasts are verified against the observed streamflow instead of the pseudo observations from the reference simulation, we do see a more pronounced overconfidence of the forecasts as well, especially at short lead times. We plan to mention this in the revised manuscript.

In addition, we will give more detailed answers to all your comments during the upcoming revision process and include your suggestions which we think will clearly improve the manuscript.

References: Monhart, S., Spirig, C., Bhend, J., Bogner, K., Schär, C. and Liniger, M. A.: Skill of Sub-seasonal Forecasts in Europe: Effect of Bias Correction and Downscaling using Surface Observations, J. Geophys. Res. Atmos., 1–18, doi:10.1029/2017JD027923, 2018.

---

## Author Response (AR1)

**Interactive discussion on «Subseasonal hydrometeorological ensemble predictions in small- to medium-size mountainous catchments: Benefits of the NWP approach» by Monhart et al.**

*Anonymous Referee #1*

*Review on the paper by Samuel Monhart et al. Subseasonal hydrometeorological ensemble predictions in small- and medium-size mountainous catchments: Benefits of the NWP approach Presented for the review is a paper dedicated to the evaluation of subseasonal streamflow forecasts performance in three mountainous catchments in Switzerland produced by the two approaches. Both approaches involve a regional hydrological process-based model PREVAH to account for the initial conditions in the catchment under consideration and the main concern of the paper is concentrated in the model driving for the forecast lead-times. The first approach is the Ensemble Streamflow Prediction (ESP) framework as described by Day (1985), which uses the historical weather data to force the model for the forecast lead-time resulting in the ensemble of the streamflow hydrographs. The second approach is the Numerical Weather Prediction (NWP) framework that involves a meteorological large-domain model-based 5-member forecasts by the ECMWF IFS to force the hydrological model for the lead-time period. The authors use both the raw and bias-corrected NWP forecasts in terms of meteorological and hydrological forecasting skill. The performance of the forecasting approaches is evaluated both for deterministic and probabilistic properties, e.g. the average characteristics are benchmarked by the MAE, NSE and NSElog criteria and the ensemble spread is evaluated by CRPSS metric, as well as the forecast spread to error ratio; the reliability of the forecasts is further examined by constructing the rank histograms. The overall importance of the study is crucial beyond doubt, as is very well described in the Introduction section – the ensemble forecasting methodology is now employed in many forecasting centers around the globe, yet the mentioned improvement in the NWP systems that hydrological prediction systems may benefit from is achieved mainly in Europe and North America, where the outstanding effort to it is applied. The case study catchments choice matches the research aims very well, as very diverse streamflow generation conditions are within the scope of the study – snowmelt-driven and fast-responding catchments are considered, which are an effort in constructing a well-performing streamflow model, as well as reliable subseasonal forecast, especially for summer and fall rainy periods with short hydrological system memory. Still, the authors show good model evaluation metrics. The main findings in the paper are in different effects of NWP bias-correction on the forecast performance, which vary in terms of variable, space and time, e.g. pre-processing of the input forcing is evaluated for temperature and precipitation apart and combined, and discussed for the three catchments over several seasons. The forecasts performance is evaluated not only for the streamflow but for the snow water equivalent in the catchments, as well, which is very crucial for the understanding of the predictability of snowmelt runoff. The results show the NWP pre-processed temperature forecasts outperforming the ESP forecasts, which is a crucial finding, as well.*

*My overall perception of the paper is that it presents an outstanding scientific effort, which is of critical importance to the modern hydrological forecasting systems research. The motivation is well described, the methods are concise and well referenced, the results are well documented and discussed and a number of very crucial statements on the topic are made. I would recommend minor revisions before the paper can be published.*

*Specific replies:*

*Anonymous Referee #1*
*A few concerns that I would appreciate the authors to enhance in the paper.*
*First is the statement in section 3.2 concerning the minor importance of such variables as*
*relative humidity etc. on the model performance on such timescales. Hence, the authors state*
*that the relative humidity values were taken from the forecasts without any pre-processing. I*
*would not agree with the authors on the minor importance of the relative humidity, as the*
*evaporation rate is highly dependent on the relative humidity, especially within the processes*
*of evaporation from snow. Given that the temperature forecasts are pre-processed,*
*the close relation of the relative humidity and temperature may influence the forecast*
*performance. However, the assessment of this was beyond the scope of the study.*

**Reply:**
As you mention, variables such as relative humidity play a crucial role regarding the evaporation rates. With our statement we do not intend to undervalue the effect of relative humidity on the hydrological forecast. We rather aim at highlighting that uncertainties at the subseasonal timescale which are already large in case of precipitation and temperature forecasts and their effect on the streamflow has not yet been investigated for the setup presented in this analysis. Hence, we believe that not pre-processing relative humidity (and other variables like sunshine duration) is justified within this analysis but should be further investigated in future studies. However, this will be a challenging task and it might be necessary to choose a different approach, as observational data at the grid scale is at least to date not available in Switzerland.
Hence, we agree with your comments and reformulated the corresponding paragraph by mentioning the importance the additional variables and pointing out that a conclusive analysis of pre-processing additional variables such as relative humidity is be beyond the scope of this study.

To account for this comment, we decide to weaken the statement in the revised manuscript.

We replaced the sentence "In addition, these variables are of minor importance for the forecst time scale investigated in this study" with the following sentence:

P7; L4-6
"Although these parameters could influence the hydrological relevant processes, e.g. evaporation rates from snow based on the relative humidity, a thorough assessment of the effect of bias correcting and downscaling of these additional variables is out of scope of the current study."

*Anonymous Referee #1*
*In the SWE forecasts verification section 4.3.4 the forecasts are verified against the*
*reference model run instead of the actual observation, yet I would appreciate if the*
*reference model performance could be discussed at least within a few sentences. Another*
*consideration is that the methodology of the SWE assessment should be placed*
*in the corresponding subsection within section 2.*

**Reply:**
We accounted for this by including a brief discussion of the findings presented in Jörg-Hess et al. (2014) where they found good agreement between the predicted SWE and the observed SWE maps for the entire 32 day forecast horizon.

The new sentence read:

P12; L 3-5
"A verification of modelled SWE against a consistent and homogenized climatology of gridded SWE based on station information is given by Jörg-Hess et al. (2014). They have shown that the modelled SWE exhibit errors that are in the same order as natural variability. "

In addition, we now mention in section 2 that the verification is performed for both variables, streamflow and SWE.

P5; L17-18
The verification is performed for the two variables streamflow and snow water equivalent (SWE).

*Anonymous Referee #1*

*Anonymous Referee #1*
*Minor technical note: p. 13 l. 1 -*
        *"...seasonal meteorological..." must be followed with a noun, which is missing*

**Reply:**
Done: "…seasonal meteorological predictions…"

*Anonymous Referee #2*

*General comments This is an interesting and thorough assessment of an ensemble streamflow forecasting system in snow-affected mountainous catchments. The system pairs NWP forecasts with a distributed hydrological model that includes detailed accounting of cryogenic processes. The system is technically advanced and in my opinion of high interest to the readership of HESS. The study is well conceived and very clearly written. A particular strength of the study is the extensive and thorough verification of the forecasts, encompassing multiple appropriate measures of performance that are described and discussed in clear and interesting ways. In general, the authors' conclusions are strongly supported by their analyses. I have one quibble with the use of QM as a means for downscaling meteorological forecasts, which amounts to a minor revision. Other than this, I have no hesitation in recommending this study for publication.*

**General comments:**

*Anonymous Referee #2*
*It appears that the authors use QM to downscale NWP predictions from a coarse grid (_30/60 k) to a fine grid (_2 k). It's well established that QM is not theoretically ideal for this practice, because of so-called variance inflation (Maraun 2013). The authors appear to be aware of this, as they discuss this issue in an accompanying paper (Monhart et al. 2018). However, it is more salient in this paper, because of the hydrological modelling that is carried out. Variance inflation is only an issue when quantile mapped/downscaled meteorological forecasts are spatially reaggregated, which is exactly what the hydrological model does. So it will not show up in the analyses carried out by Monhart et al. 2018 (where variables are not reaggregated), but it could well be an issue in this study.*

*In addition, and as Maraun shows, the variance inflation problem is only strongly evident for extremes. Extremes are not the focus of the analysis carried out in this paper, which is fine. But this means it's hard to tell if variance inflation is present in streamflow forecasts. As the forecasting system could be used for flood prediction, this may be a serious issue. Accordingly, I recommend two changes to the manuscript:*

*1) The authors should clearly describe how they bridge the gap in spatial resolution from a _30/60 k horizontal grid (NWP forecasts) to a _2 k horizontal grid (observations).*

**Reply:**
We now included a more concise description on how the gap between the coarse spatial resolution of the NWP predictions and the 2 km gridded observation is performed. To account for this concern, we expanded subsection 2.1 where the pre-processing step using QM is described.

P4; L10-15
"The pre-processing is performed for temperature and precipitation and involves not only a bias correction but also a downscaling because of the higher resolution of the gridded observation data used in this study. The observation and forecast data used in this study is described in more detail in the Section 3. However, it is worth mentioning here that the raw model resolution of 50 km is bias corrected with QM using gridded observations with a

higher spatial resolution of 2km. This resolution corresponds to the meteorological input of the hydrological model, for which observations from station data are interpolated to 2 km grids (see section 3.2)."

*Anonymous Referee #2*
*And if, as I've assumed, they use QM for this purpose:*
*2) The authors should briefly acknowledge the issue of variance inflation in the discussion, including a discussion of possible implications for their system (perhaps alongside recommendations for dealing with these implications).*

**Reply:**
As you correctly assume (and as we already answered in the short reply) we do use QM to downscale the predictions and we are aware of the variance inflation issue caused by this method. We now included a new paragraph in the discussion to highlight the limitations of QM related to the variance inflation. We acknowledge that the problem of variance inflation can influence the results and argue why we believe it is still justified to use QM for the downscaling within our study. As you suggested we now include reference that discuss the problem in more detail and propose alternative approaches that could be used in future studies. The added paragraph reads as follows:

P13; L7-L21
"An additional critical limitation of QM is the issue of variance inflation. Maraun ( 2013) emphasizes that the variance of the downscaled product strongly depends on the variance of the raw model grid box and QM does not introduce any small-scale variability. This is of particular importance for applications using local-scale information (such as distributed hydrological modelling) and if extremes are considered. In CH2018 (2018) these limitations of the QM method are highlighted for local climate change scenarios in Switzerland. In particular for convective precipitation events in summer the variance inflation issue can cause misinterpretation of data at the finer resolved scale. In the present study we are interested in the average streamflow throughout all season in the year for the upcoming 32 days and not in predicting extremes what reduces these implications, but still the spatial structure especially during convective situations in summer will likely be misrepresented and can influence the results. Different alternatives could be used depending on the specific application of the downscaled information, e.g. perfect prognosis approaches (Von Storch, 1999), the use of weather generators (e.g. Peleg et al., 2017) or in general stochastic methods (e.g. Volosciuk et al., 2017). However, such methods often require large computational resources. As the intention of this study is to pioneer the use of subseasonal hydrological predictions towards an operational use, we decided to use the QM technique despite its limitations. The results presented above and discussed in the following paragraphs legitimate our choice. Nevertheless, future studies should focus on the effect of variance inflation when QM is used to pre-process the predictions and alternative methods should be considered."

*Anonymous Referee #2*
*I also encourage the authors to consider measuring the impact of variance inflation on their system in future work.*

**Reply:**
We will consider this issue in our future work by either choosing a different correction technique and/or a careful assessment of the effect resulting from the inflation issue.

*Anonymous Referee #2*
*Specific comments*
*Page 4 L7-9 "For a given target day of a reforecast the correction is derived from the
distribution of all the reforecasts within a three weeks window around the same lead day and
the corresponding observations, hence the correction depends both on the lead time and on
the period of the years".*
*Is the QM cross-validated in some way?*

**Reply:**
Yes, the forecasts are cross-validated. The calibration of the hindcasts is performed in a
leave-one-year-out cross-calibration framework. Hence in the verification the information of
the year to be verified is not used in the calibration procedure i.e. cross-validated. We added
the following statement to make that clear.

P4; L8-9
        "This cross-calibration framework ensures a cross-validation described in subsection
2.3."

*Anonymous Referee #2*
*How are zero values in precipitation handled in the QM?*

**Reply:**
Zero values are not handled in a specific way. Because we do apply a multiplicative
correction zero values will not cause an issue. Therefore, the QM does not artificially produce
rain, i.e. in case zero precipitation is forecasted the corrected precipitation still has zero
precipitation. In general, weather prediction models exhibit a drizzle effect due to their large
grid sizes, meaning that raw model forecast generally predict too much rain compare to the
observation (in case of very low precipitation rates). Hence, zero precipitation values do not
need special treatment during the pre-processing.
We now included an additional sentence to highlight that the in the multiplicative version of
QM zero values do not need special treatment in the pre-processing step.

P 4; L20-21
        "Using the multiplicative version of QM for temperature allows to include zero value
without special treatment. Hence, no precipitation can be generated if the raw forecasts do not
exhibit any rain."

*Anonymous Referee #2*
*Page 5 Section 2.4 For each score used, please note the range of values taken and
the orientation of the score (e.g. -infinty to 1 for NSE, with 1 being perfect) This allows
easy intepretation of, e.g., Figure 4.*

**Reply:**
We now include the following statement in the figure caption.

P31, L11-13
"The NSE and the CRPSS span from -infinty to 1 with a perfect score being 1; a bias of zero
indicates no forecast error with negative values indicating underestimation and positive
values indicating overestimation of the flow; reliable forecasts exhibit a SprErr of 1 and
lower values indicate overconfidence and greater values indicate overdispersion."

*Anonymous Referee #2*
*L16 "we use the spread to error ratio (SprErr) as an indicator for the forecast reliability"*
*Please briefly describe how this is calculated.*

**Reply:**
We now added a sentence to describe how the spread to error ratio is calculated:

P5; L29-32
        "The *SprErr* is defined as the ratio between the variance of the forecast ensemble (forecast spread) and the mean squared error (MSE) of the ensemble forecast (forecast error). For reliable forecasts the spread and the error are equal, resulting in a SprErr of 1 whereas values below 1 indicate overconfidence (errors are larger compared to the spread) and values above 1 indicate overdispersion (the spread is larger compared to the error)."

*Anonymous Referee #2*
*Page 8 L18 Figure 3. It's very difficult to see the different colours in this figure, especially in the right hand panel - i.e., it's not possible to distinguish QM from raw. Please replot so this is clearer (e.g., with different colours/box outlines, and/or perhaps restrict the vertical axis in the rh panel to [-0.2 0.3]).*

**Reply:**
We change the figure according to your suggestions.

*Anonymous Referee #2*
*L22-23 "After bias correction the skill is higher with positive CRPSS up to three weeks in winter and spring." From the figure, precip skill looks to be negligible in DJF after week 1. Skill scores will of course be a little noisy (in time, as well as in space; the authors have only considered spatial variation), so I don't think the authors should describe forecasts as 'skillful' if they have CRPSS values only very slightly above zero.*

**Reply:**
We agree, the description is too optimistic. We considered this comment and only mention spring in this sentence. It now reads:

P9; L19-20
"After bias correction the skill is higher with positive CRPSS up to three weeks in MAM. In JJA the positive skill is only observed for 2 weeks lead time and in SON and DJF for the first week only."

*Anonymous Referee #2*
*L28 NSE, Bias - I assume these are calculated on the mean of the ensemble? Please state this in Section 2.4.*

**Reply:**
Your assumption is correct. This will be included in Section 2.4

Page 5; L23-24:
"For both versions of the NSE and the bias the ensemble mean is used for the calculation. "

*Anonymous Referee #2*
*Page 9*
*L7-8 "The negative biases of the ESP approach indicate an underestimation of*
*the streamflows for all lead times in the Verzasca catchment." It's not clear to me why*
*ESP predictions would be biased. ESP forcings, by construction, are unbiased. As the*
*bias in predictions is calculated against model climatology, there should be no bias, as*
*occurs in the Thur. Please briefly explain what is going on here.*

**Reply:**
We generally agree with your statement that ESP predictions are by construction unbiased.
The historical meteorological observations used to run the hydrological simulations are a
sample of the climatology and thus the resulting streamflow prediction should theoretically
agree (i.e. be unbiased) with the climatological streamflow used for the verification.
However, there are two effects that might lead to biases in the ESP prediction. First, if the
initial conditions at the time of the forecast initialization strongly deviates from the
climatology the ESP prediction will take more time until the streamflow converges with
climatology, in particular in snow-dominated catchments as the Verzasca and the Klöntal
catchment. Second, the meteorological input for the ESP predictions in our case is not an
exact sample of the climatology. The meteorological observations from 1980 to 2014 are
used, the streamflow climatology is based on the period 1994-2015. Hence, trends in the
meteorological input might affect the streamflow prediction resulting in a bias. E.g.
temperature exhibits a positive trend within this period, colder temperature in the beginning
of the period might lead to negative biases (underestimation of the streamflow) because the
storage of precipitation in snow is enhanced. Again, this effect is stronger in snow-dominated
catchments.

*Anonymous Referee #2*
*L15 "The spread error ratio of the ESP predictions is below 1 for all lead times indicating*
*overconfidence." There are two issues here. First, as I already alluded to, it would be*
*beneficial to readers unfamiliar with the spread-error ratio to offer a brief explanation of*
*the range of values it can take, and which direction indicates over/under confidence in*
*Section 2.4.*

**Reply:**
We now adapted the description in the methodology.

P5, L30-32:
"For reliable forecasts the spread and the error are equal, resulting in a SprErr of 1 whereas
values below 1 indicate overconfidence (errors are larger compared to the spread) and values
above 1 indicate overdispersion (the spread is larger compared to the error)."

*Anonymous Referee #2*
*Second, I can't understand why the ESP forecasts are not reliable. ESP*
*forcings are by construction reliable, so the spread-error ratio for the ESP forecasts*
*should be close to 1. But this is not so in the two smaller catchments. Why is this?*

**Reply:**
We suggest a similar explanation as in the answer to the comment related to the bias of the
ESP (first answer on this page). In short, different time period of the meteorological forcing

of the ESP (1980-2014) and the streamflow climatology (1994-2015) might affect the SprErr as well. The forecast in the longer period might lead to larger error what could lead to a reduced SprErr. In addition, the effect of initial conditions that are strongly deviating from the climatological state can lead to larger errors especially at early lead times until the predictions converge to the climatology.

*Anonymous Referee #2*
*Page 10*
*L26-27 "The rank histograms for the ESP predictions do provide more uniform*
*rank histograms with a weak tendency of a negative bias." Again, I would be interested*
*in a brief explanation of this bias in ESP forecasts.*

**Reply:**
See answer to the comment above.

*Anonymous Referee #2*
*Page 12*
*L15 "QM indeed is able to provide reliable ensembles" To me this sounds*
*as though QM is responsible for the reliable ensembles, and this isn't really correct.*
*QM can improve reliability to the extent that it is negatively impacted by bias (including*
*conditional bias - i.e., biases at different points in the marginal distribution). You*
*have shown (like Zhao et al.) that QM does not correct for overconfidence in short*
*lead-time forecasts. This is because the underlying forecasts are overconfident - QM*
*can't correct this (by construction). At longer lead-times, the QM forecasts are reliable*
*because the spread in the underlying forecasts is appropriate (notwithstanding conditional*
*biases). The same goes for coherence (discussed in the accompanying Monhart*
*et al. 2018 JGR paper evaluating the NWP forecasts) - QM is not capable of correcting*
*negative skill wrt to climatology in forecasts, other than that due to bias (again, by*
*construction). The forecasts presented in this study are coherent because the underlying*
*(raw) forecasts appear to be neutrally skillful at long time scales. In other words,*
*it is the combination of NWP model and the QM that create the reliable and coherent*
*forecasts shown in this study, not just QM. I think it would be better to reword what's*
*written here to reflect this fact.*

**Reply:**
We account for this comment and reworded the statement to make it clear that QM alone is not able to provide reliable forecasts if the underlying meteorological forecast are not reliable.
The sentence now reads:

P13; L4-7
"To summarize, it was found that the combination of the NWP model with QM indeed is able to provide reliable ensembles for lead times beyond 10 days but at shorter lead times the ensembles tend to be overconfident because the spread in the underlying NWP forecasts tends to be inappropriate what cannot be corrected using QM."

*Anonymous Referee #2*
*P14*
*L9-10 "to correct the errors and biases of the hydrological simulation" and, presumably,*
*to account for additional uncertainty induced by the hydrological model in the ensemble?*

**Reply:**
Thanks for this suggestion. We included this, and the sentence now reads:

P15; L32-33
"…to correct the errors and biases of the hydrological simulation and to account for additional uncertainty induced by the hydrological model in the ensemble."

*Anonymous Referee #2*
*P14 "To do so we verified the streamflows of the reference simulation of the PREVAH model against observations." One thing not discussed here is reliability. I assume when assessed against observations, the ensembles are highly overconfident because uncertainty in the hydrological model is not included in the ensemble (see, e.g., Bennett et al. 2014). This is especially true at very short lead times (perhaps <3 days), when hydrological model uncertainty may be the dominant source of uncertainty in the forecasts. This may be worth mentioning.*

**Reply:**
Here we discuss the performance of the reference simulation verified against observations. The reference simulation is a single timeseries and thus no ensembles can be verified. But for the ensemble predictions verified against the observation we indeed find a pronounced overconfidence at short lead times. Hence the results confirm what you mention in your comment. Although we decide not to include any additional figures we mention this aspect in the discussion.

P16, L13-L18:
"An example of such a deficiency is the uncertainty resulting from the hydrological modelling that result in stronger overconfidence especially at short lead times when the hydrological model uncertainty may be the dominant source of uncertainty as discussed for example in Bennett et al., (2014). To illustrate this for the prediction used in this study, the verification was repeated with the real runoff observations. The skill in most seasons (MAM, JJA, SON) exhibit the same behavior with high skill at early lead times and decreasing skill at longer lead times and generally higher overconfidence at short lead times is observed what confirms the findings by Bennett et al. (2014)."

*Typos/Grammar Page*
*1 L14 "Prior of" should be "Prior to" or more simply "Before"*
Done

*Page 2 L11 "both," delete comma*
Done

*Page 5*
*L2 "comparison of to" delete "of"*
Done

*L28 "year" should be "years"*
Done

*Page 7 L3 "station" should be "stations"*

Done

*Page 8 L9 "evaluate of operational" delete "of"*
Done

*Page 9 L9 "and reach" should be "and reaches"*
Done

*L11 "enhance" should be "enhances"*
Done

*L11 "elongates positive up" I think "skill" is missing here - i.e. "elongates positive skill up"*
Done

*L27 "are shown" should be "is shown"*
Done

*Page 13 L1 "in seasonal meteorological can" I think this should be "in seasonal meteorological forecasts can"*
Done

*Page 14 L11 "prediction" should be "predictions"*
Done

*L23 "exhibit" should be "exhibits"*
Done

*Page 15 L14 "enhance" should be "enhances"*
Done

*L24 "EPS" should be "ESP"*
Done

*L25 "both, temperature" delete comma*
Done

*Page 27 L5 "whereas" should be "where"*
Done

*References Bennett JC, Robertson DE, Shrestha DL, Wang QJ, Enever D, Hapuarachchi P, Tuteja NK. 2014. A system for continuous hydrological ensemble fore-casting (SCHEF) to lead times of 9 days. Journal of Hydrology 519: 2832-2846. DOI: 10.1016/j.jhydrol.2014.08.010.*
*Monhart, S., Spirig, C., Bhend, J., Bogner, K., Schär, C. and Liniger, M. A.: Skill of Sub-seasonal Forecasts in Europe: Effect of Bias Correction and Downscaling using Surface Observations, J. Geophys. Res. Atmos., 1–18, doi:10.1029/2017JD027923, 2018.*

*Zhao, T., Bennett, J. C., Wang, Q. J., Schepen, A., Wood, A. W., Robertson, D. E. and Ramos, M. H.: How suitable is quantile mapping for postprocessing GCM precipitation forecasts?, J. Clim., 30(9), 3185–3196, doi:10.1175/JCLI-D-16-0652.1, 2017.*
*Interactive comment on Hydrol. Earth Syst. Sci. Discuss., https://doi.org/10.5194/hess-2018-458, 2018.*

Anonymous Referee #3
*General comments.*

*The main motivation of this study is to fill the gap in small scale researches in determination of the propagation positive skill extent in meteorological prediction models further into the streamflow forecasts. To address this problem, a traditional ESP approach was compared with prediction driven by ECMWF subseasonal ensemble system in three alpine catchments with varying hydroclimatic conditions. To emphasis the effect of applying pre-processing (QM-based) of NWP output, prediction verification was done against the reference simulation (pseudo observations). Thus hydrological model errors were excluded from the analysis.*
*Summary.*
*There was in-depth discussion on hydroclimatic variability and predictability, the role of forcing and model parameters' uncertainty. The verification metrics used were relevant and applied in a logical manner. The results well supported the conclusions. Some sections recommend recompiling for better logically organized and easy follow.*

*I recommend publishing the manuscript but encourage the authors:*

*1) Give a more justification on choice of these watersheds for sub-seasonal forecasting. Initially, it can be supposed that the study is a part of a large numerical experiment where the results were confirmed only for the three arbitrary watersheds.*

**Reply:**
The choice of the watersheds was driven by the application of the forecast and the diversity in a hydroclimatic sense. Within the larger umbrella project, we aim at analysing and quantifying the benefit of using subseasonal forecasts for the optimization of hydropower operations to increase their revenues in Switzerland. Therefore, watersheds with installed hydropower operations were selected first (the Klöntal and the Verzasca catchment, representing snow dominated and partially glaciated catchments). To broaden the scientific value of our analysis and better quantify the effect of snow, we included as well the Thur catchment which is precipitation dominated. Hence, the watersheds are not an arbitrary choice from a large numerical experiment but rather a reasonable compromise between further use of the results for our intendent application and a scientific in-depth analysis of the forecast performance considering different hydroclimatic regimes.
We account for this by explaining the motivation for the choice of the watersheds in section 3.3 of the analysis.

P7; L10-15
"The selection of the catchments is a compromise between the intended applications of our results within hydropower optimization in the Alpine region and to adequately meet the requirements of a scientific analysis. Hence, the Klöntal and the Verzasca catchment both are selected because of existing hydropower installation in these watersheds and the Thur catchment was chosen as a representative catchment with different hydroclimatic characteristics and because the catchment is often considered in hydrological research in Switzerland."

*Anonymous Referee #3*
*2) Add meteorological observations network on fig. 2 and give some comments explaining the good modeling quality (tab. 1) when using the grid product obtained at a low observations network density.*

**Reply:**
As mentioned in the manuscript, the grid product is often used in climate related research in Switzerland. The relevant publication on the gridded product, including verification studies of the gridded product are given in the manuscript. An overlay of the station used to produce the gridded data set is in our opinion an over overkill for the present study as the information can be found in in the given references. Therefore, we decide not to include the observational stations in figure 2. The good modelling quality, speaking of the performance of the streamflow predictions indicate that the gridded product does provide a good baseline for pre-processing subseasonal hydrological predictions.

We included the following sentence in the discussion:
P14, L1-3:
"Furthermore, the improvements in performance of the streamflow predictions by pre-processing suggests that the gridded observational dataset provide a good baseline for this purpose, despite the difficulties involved in producing a gridded product based on a limited number of observational stations."

*Anonymous Referee #3*
*3) Give a number of predictions made for evaluation.*

**Reply:**
For the verification we use a total number of 1040 reforecasts with 5 members each. This number results from the dataset which provides 1 reforecast per week for the 20-year period (1994-2014). For the seasonal aggregation a total of 260 reforecasts is used for the verification.

We added this in the section 3.1. the sentence now reads:

P6; L20-21
"… covering the period from April 1994 to March 2014 resulting in a total of 1040 individual reforecasts that are analyzed within this study."

*Anonymous Referee #3*
*4) Comment on how the processing of only temperature and precipitation affects and propagate through the hydrological simulation.*

**Reply:**
The results clearly show that pre-processing both temperature and precipitation is important to provide well performing hydrological simulation. Hence, pre-processing temperature only does have a large impact on the performance of the streamflow predictions but if precipitation is pre-processed as well, the rank histograms indicate more reliable forecasts. This subject is addressed in the results section 4.3.

*Anonymous Referee #3*
*5) Specify if the ESP method can outperform the NWP if ensemble takes not all but only individual years guided by a certain criterion for the similarity of the initial conditions.*

**Reply:**
As you mention, the performance of the ESP predictions could be enhanced based on a selection of years where the initial conditions show high similarity with present conditions. E.g. Crochemore et al. (2017) showed that seasonal forecasts can benefit by conditioning climatology. However, a thorough assessment of conditioning the ESP predictions is beyond the scope of the present study but could be considered for further studies within this area.

We widened the discussion to make the reader aware of this potential.

P14, L4-7:
"In addition, the performance of the ESP predictions could potentially be enhanced if not all, but only individual years are taken into account. A certain guidance based on a selection of years with similar initial conditions could be taken into account. Crochemore et al. (2017) have shown that seasonal prediction based on ESP can benefit from condition the forecasts on climatology. However, an evaluation of such an approach is out of scope of the present study."

*Anonymous Referee #3*
6) Recompile the sections 2 and 3 referred to each other to make them more consistent.

**Reply:**
As a result of including the comments of the all three reviewers, some changes have been made to section 2 and 3 as well. Both sections should now be more consistent.

Specific comments.
P.13, L5. The upper (reads like nested) Thur subcatchment Halden (1750 km2) is little bit bigger then Thur watershed itself (1696 km2).

**Reply:**
You are right, the Thur subcatchment Halden is 1085 km2. We changed this accordingly.

[revised manuscript text omitted]